# BECAUSE:
# Bilinear Causal Representation for Generalizable Offline Model-based Reinforcement Learning

**Haohong Lin[1], Wenhao Ding[1], Jian Chen[1], Laixi Shi[2], Jiacheng Zhu[3], Bo Li[4], Ding Zhao[1]**
[1]CMU,   [2]Caltech,   [3]MIT,   [4]UChicago & UIUC
{haohongl, wenhaod}@andrew.cmu.edu

## Abstract

Offline model-based reinforcement learning (MBRL) enhances data efficiency by utilizing pre-collected datasets to learn models and policies, especially in scenarios where exploration is costly or infeasible. Nevertheless, its performance often suffers from the objective mismatch between model and policy learning, resulting in inferior performance despite accurate model predictions. This paper first identifies the primary source of this mismatch comes from the underlying confounders present in offline data for MBRL. Subsequently, we introduce **B**ilin**E**ar **CAUS**al r**E**presentation (BECAUSE), an algorithm to capture causal representation for both states and actions to reduce the influence of the distribution shift, thus mitigating the objective mismatch problem. Comprehensive evaluations on 18 tasks that vary in data quality and environment context demonstrate the superior performance of BECAUSE over existing offline RL algorithms. We show the generalizability and robustness of BECAUSE under fewer samples or larger numbers of confounders. Additionally, we offer theoretical analysis of BECAUSE to prove its error bound and sample efficiency when integrating causal representation into offline MBRL. See more details in our project page: https://sites.google.com/view/be-cause.

## 1 Introduction

Offline Reinforcement Learning (RL) has shown great promise in learning directly from pre-collected datasets, especially in scenarios where active interaction is expensive or infeasible [1]. Specifically, offline model-based reinforcement learning (MBRL) [2, 3, 4], learning policies with an estimated world model, generally perform better than their model-free counterparts in long-horizon tasks such as self-driving vehicles [5], robotics [6], and healthcare [7]. However, offline RL suffers from distribution shift because the rollout data is either sampled from some suboptimal behavior policies or sampled from slightly different training environments compared to the deployment time [8].

Although identifying distribution shift issues, many of the current offline MBRL works fail to model the shift in environment dynamics, which is ubiquitous and could cause catastrophic failure of trained policy at a slightly different deployment stage. Furthermore, since the learning objectives of the world models and policies are isolated from each other, a significant challenge in offline MBRL is objective mismatch [9, 10] problem (shown in Figure 1): models that achieve a lower training loss are not necessarily better for control performance. For example, a dynamics model achieve relatively low

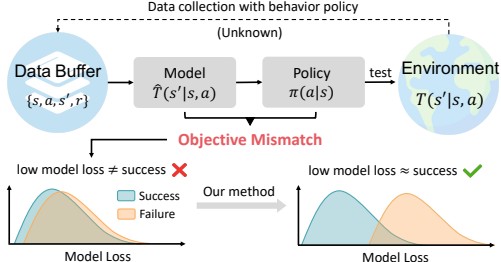

Figure 1: The objective mismatch problem.

38th Conference on Neural Information Processing Systems (NeurIPS 2024).

prediction loss, yet it may not be sufficient to guide the planner to the high-reward region. Previous works [9, 10] have attempted to reduce such objective mismatch by jointly learning the model and policy. However, the performance is suboptimal due to the lack of digging into the underlying cause of the mismatch issue [8].

In this work, we identify that the objective mismatch between model estimation and policy learning comes from two sources of *distribution shift* in offline MBRL: (1) shift between the online optimal policy and offline sub-optimal behavior policies, and (2) shift between the data collection environment and online testing environments. Unlike humans, who make decisions based on reasoning over task-relevant factors, models in offline RL memorize correlations without learning the causality. The sub-optimal behavior policies introduce *spurious correlations* [11] between actions and states, making the model memorize specific actions. When online testing environments differ from the data collection environment, the model could overfit spurious correlations in the state and fail to generalize to unseen states. Based on the analysis of the above mismatch, our work differs from previous work of causal model-based RL [12, 13] in that we model causality in both model and policy learning., We aim to avoid spurious correlation by discovering underlying structures between abstracted states and actions.

To alleviate objective mismatch and generalize well, we introduce the BilinEar CAUSal rEpresentation (BECAUSE) that integrates the causal representation in both world model learning and planning of MBRL agents. Inspired by preliminary works that use bilinear MDPs to capture the structural representation in MBRL [14], we first approximate the causal representation to capture the low-rank structure in the world model, then use this learned representation to facilitate planning by quantifying the uncertainty of sampled transition pairs. Consequently, we factorize the spurious correlations and learn a unified representation for both the world model and planner.

In summary, the contribution of this paper is threefold:

- We formulate offline MBRL into the causal representation learning problem, highlighting the tight connection between structural causal models and low-rank structures in MDPs. To the best of our knowledge, this is the first work that systematically reveals the connection between causal representation learning and Bilinear MDPs.

- We propose BECAUSE, an empirical causal representation framework, based on the above formulation. BECAUSE first learns a causal world model, then fosters the generalizability of offline RL agents by quantifying the uncertainty of the state transition, which facilitates conservative planning to mitigate the objective mismatch.

- We provide extensive empirical studies and performance analysis in tasks of multiple domains to demonstrate the superiority of BECAUSE over existing baselines, which illustrates its potential to improve the generalizability and robustness of offline MBRL algorithms.

## 2   Problem Formulation

To alleviate the objective mismatch problem and the degraded performance caused by the spurious correlation, we first provide our novel formulation of learning the underlying causal structures of Markov Decision Process (MDP) under the bilinear MDP setting, then introduce the causal discovery for MDP with confounders.

### 2.1   Preliminary: MDP and Bilnear MDP

We denote an episodic finite-horizon MDP by $\mathcal{M} = \{\mathcal{S}, \mathcal{A}, \mathcal{T}, H, r\}$, which is composed of state space $\mathcal{S}$, action space $\mathcal{A}$, a set of transition functions $\mathcal{T}$, planning horizon $H$ and reward function $r$ associated with task preferences. Without loss of generality in many real-world practices, we assume that the reward function is bounded by $r_h \in [0, 1], \forall h \in [H]$. Specifically, we are interested in a goal-conditioned reward setting, where $\forall g \in \mathcal{S}, r(s, a; g) = 1$ if and only if $s = g$.

Given a policy $\pi$ and the state-action pair $(s, a) \in \mathcal{S} \times \mathcal{A}$, we then define the state-action value function in the timestep $h$ as $Q_h^\pi(s, a) = \mathbb{E}_\pi \left[ \sum_{i=h}^{H} r_i(s_i, a_i) | s_h = s, a_h = a \right]$, and the value function $V_h^\pi(s) = \mathbb{E}_\pi \left[ \sum_{i=h}^{H} r_i(s_i, a_i) | s_h = s \right]$. The expectation $\mathbb{E}_\pi$ here is integrated into randomness throughout the trajectory, which is essentially induced by the random action of the policy $a_i \sim \pi(\cdot|s_i)$

and the time-homogeneous transition dynamics of the environment $s_{i+1} \sim T(\cdot|s_i, a_i), \forall i \in [h, H]$. In the offline dataset, the data rollouts can be seen as generated by some (mixed) behavior policy $\pi_\beta$, resulting in a dataset $\mathcal{D}$ with in total $n$ samples $\{s_i, a_i, s_i', r_i\}_{1 \leq i \leq n}$.

**Definition 1** (Bilinear MDP [14]). For each $(s, a) \in \mathcal{S} \times \mathcal{A}, s' \in \mathcal{S}$, we have the corresponding feature vector $\phi(\cdot, \cdot) : \mathbb{R}^{|S|} \times \mathbb{R}^{|A|} \to \mathbb{R}^d, \mu(\cdot) : \mathbb{R}^{|\mathcal{S}|} \to \mathbb{R}^{d'}$. With some core matrix $M \in \mathbb{R}^{d \times d'}$, we can represent the transition function kernel $T(\cdot|\cdot, \cdot)$ as

$$\forall s, a, s' \in \mathcal{S} \times \mathcal{A} \times \mathcal{S}, \ T(s'|s, a) = \phi(s, a)^T M \mu(s'), \tag{1}$$

where $\phi(s, a)$ and $\mu(s')$ are embedding functions that map the original state and action to the latent space, $M$ is the core matrix that models the transition relationship between the previous timestep and next timestep in the latent space. Such a linear decomposition in the transition dynamics allows us to embed structures of the transition model without the loss of general function approximation capabilities to derive state and action representations.

## 2.2 Action State Confounded MDP

We consider the existence of confounders in the MDP to represent the offline data collection process, and define action-state confounded MDP (ASC-MDP):

**Definition 2** (ASC-MDP). Besides the components in standard MDPs $\mathcal{M} = \{\mathcal{S}, \mathcal{A}, T, H, r\}$, we introduce a set of unobserved confounders $u$. In ASP-MDP, confounders are factorized as $u = \{u_\pi, u_c\}_{1 \leq h \leq H}$, where $u_\pi \in \mathcal{U}$ denotes the confounders between $s$ and $a \sim \pi_\beta(s)$ induced by behavior policies, and $u_c \in \mathcal{U}$ denotes the confounders within the state-action pairs of the environment transition, that is, the inherent structure between $(s, a)$ and $s'$. Here we assume a time-invariant confounder distribution $u \sim P_u(\cdot), \forall h \in [H]$, which is a common assumption [15, 16, 17]

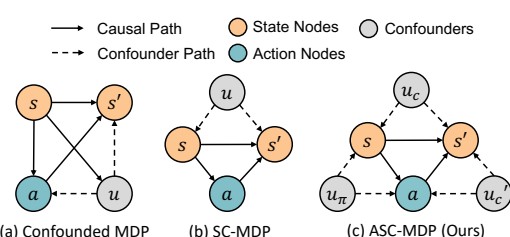

Figure 2: Comparison of our ASC-MDP with two existing formulations.

The resulting causal relationship of ASC-MDP is demonstrated in Figure 2. Originating from the original MDP, ASC-MDP is different from the Confounded MDP [18] and State-Confounded MDP (SC-MDP) [19] in that it models both the *spurious correlation* between the current state $s$ and the current action $a$, as well as those between the next state $s'$ and $(s, a)$. Yet, confounded MDP and SC-MDP only model part of the possible confounders between states and actions. The factorization of the confounder in ASC-MDP aligns with the source of spurious correlation in offline MBRL.

# 3 Proposed Method: BECAUSE

We propose BECAUSE, our core methodology for modeling, learning, and applying our causal representations for generalizable offline MBRL. Section 3.1 models the basic format of causal representations and analyzes their properties. Section 3.2 gives a compact way to learn the causal representation $\phi(s, a)$ and $\mu(s')$, as well as the core mask estimation $M$. Section 3.3 utilizes these learned causal representations in both world model learning and MBRL planning from offline datasets.

## 3.1 Causal Representation for ASC-MDP

In the presence of a hidden confounder $u$, we model the confounder behind the transition dynamics as a linear confounded MDP [18]:

$$T(s'|s, a, u) = \widetilde{\phi}(s, a, u)^T \mu(s'), \tag{2}$$

where $u \sim P_u(\cdot)$. Inspired by the Bilinear MDP in Definition 1, we decompose $\widetilde{\phi}(s, a, u)$ into a confounder-aware core matrix $M(u)$ and a feature mapping $\phi(s, a)$, which factorize the influence of

the confounders. Given the factorization of confounder $u = \{u_c, u_\pi\}$ in Definition 2, we derive via *d-separation* in the graphical model in Figure 2 that $s' \perp\!\!\!\perp u_\pi | \{s, a, u_c\}$. As a result, we only need to consider the confounder $u_c$ from the environment when decomposing the transition model:

$$T(s'|s, a, u) = T(s'|s, a, u_c) = \phi(s, a)^T M(u_c) \mu(s'). \tag{3}$$

**Definition 3** (Construction of causal graph $G$). In ASC-MDP, $G = \begin{bmatrix} 0^{d \times d} & M \\ 0^{d' \times d} & 0^{d' \times d'} \end{bmatrix}$. for all (sparse) core matrix $M$, the causal graph $G$ is bipartite, thus $\forall\, G, G \in$ DAG.

Definition 3 reveals the connection between the core matrix $M$ and causal graph $G$, as is formulated in the ASC-MDP. To reduce the influence of $u$ and estimate the unconfounded transition model $T(s'|s, a)$, one way is to identify the causal structures induced by the confounder $u_c$ for the transition dynamics [20]. Existing methods in differentiable causal discovery [21, 22, 23] transform causal discovery on some causal graph $G$, into a maximum likelihood estimation (MLE) with regularization:

$$\widehat{G} = \arg\max_{G \in \text{DAG}} \log p(\mathcal{D}; \phi, \mu, G) - \lambda |G| \implies \widehat{M} = \arg\max_{M} \log p(\mathcal{D}; \phi, \mu, M) - \lambda |M|, \tag{4}$$

Since in our case, $M$ is a sub-matrix of the causal graph $G$. Given the Definition 3, $G$ automatically satisfies the formulation of ASC-MDP, thus discovering $G$ is essentially estimating the sparse submatrix $M$ **without** DAG constraints: $\widehat{M} = \arg\max_M \log p(\mathcal{D}; \phi, \mu, M) - \lambda |M|$. We elaborate Definition 3 and show the relationship between core matrix $M$ and causal graph $G$ in Appendix A.3.

We also make the assumption that the sparse $G$ and $M$ remain invariant with different environment confounders in the offline training and online testing.

**Assumption 1** (Invariant causal graph). We denote the causal graph $G$ under confounder $u$ as $G(u)$. The generalization problem that we aim to solve satisfies the invariance in the causal graph $G(u_c)$, where $G(u_c) = G(u'_c), M(u_c) = M(u'_c)$, $u_c$ and $u'_c$ are the confounders in training and testing.

*Remark* 1. The assumption 1 can also be interpreted as task independence in [24], invariant state representation [25], or invariant action effect in [26]. See detailed comparison in appendix Table 4.

## 3.2 Learning Causal Representation from Offline Data

We first learn the causal world model $T(s'|s, a)$ in the presence of confounders $u$ in the offline datasets. As formulated in ASC-MDP 2, there are two sets of confounders: $u_\pi$ and $u_c$. To estimate an unconfounded transition model and remove the effect of confounder, we first remove the impact of $u_c$ which comes from the dynamics shift by estimating a batch-wise transition matrix $M(u_c)$, then we apply a reweighting formula to deconfound $u_\pi$ induced by the behavior policies and mitigate the model objective mismatch.

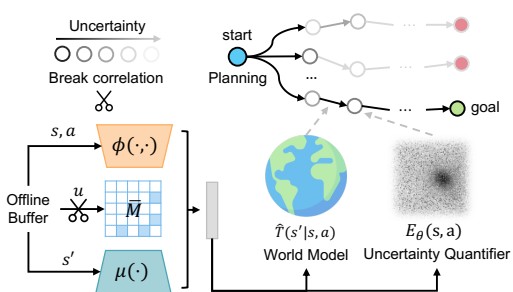

Figure 3: BECAUSE learns a causality-aware representation from the buffer and uses it in both the world model and uncertainty quantification to obtain a pessimistic planning policy.

As discussed in Definition 3, we only need to optimize the part of the parameters of the causal graph $G$, i.e. $M$. Thus, we can remove the constraints in (4), then transform the original causal discovery problem into a regularized MLE problem as follows:

$$\min_{M} \mathcal{L}_{\text{mask}}(M) = \min_{M} \left( -\log p(\mathcal{D}; \phi, \mu, M) + \lambda |M| \right)$$
$$= \min_{M} \Big( \underbrace{\mathbb{E}_{(s,a,s') \in \mathcal{D}} \| \mu^T(s') K_\mu^{-1} - \phi^T(s, a) M \|_2^2}_{\text{World Model Learning}} + \underbrace{\lambda \|M\|_0}_{\text{Sparsity Regularization}} \Big). \tag{5}$$

where $K_\mu := \sum_{s' \in \mathcal{S}} \mu(s') \mu(s')^T$ is an invertible matrix. The derivation of Equation (5) is elaborated in Appendix A.4. In practice, we use the $\chi^2$-test for discrete state and action space and the fast

Conditional Independent Test (CIT) [27] for continuous variables to estimate each entry in the core matrix $M$. We regularize the sparsity of $M$ by controlling the $p$-value threshold in CIT and provide a more detailed implementation in Appendix C.1.

Estimating the core mask provides a more accurate relationship between state and action representations, and we further refine the state action representation function $\phi$ and $\mu$ to help capture more accurate transition dynamics. We optimize them by solving the following problem, according to the transition model loss and spectral norm regularization [28] to satisfy the regularity constraints of the feature in Assumption 3:

$$\min_{\phi,\mu} \mathcal{L}_{\text{rep}}(\phi,\mu) = \min_{\phi,\mu} \mathbb{E}_{(s,a,s')\in\mathcal{D}} \|\mu^T(s')K_\mu^{-1} - \phi^T(s,a)M\|_2^2 + \lambda_\phi\|\phi\|_2 + \lambda_\mu\|\mu\|_2. \quad (6)$$

The world model learning process is illustrated in Figure 3. The estimation of individual $M(u_c)$ mitigates the spurious correlation brought by $u_c$. To further deal with the spurious correlation in $u_\pi$ induced by the behavior policy $\pi_\beta(a|s, u_\pi)$, we utilize the conditional independence property in the ASC-MDP shown in Equation (3). The following equation shows that the true unconfounded transition $T$ can be rewritten in the reweighting formulas. This reweighting process in mask $M$ serves as the soft intervention approach [18, 29] to estimate the treatment effect in the transition function $T$ in an unconfounded way:

$$
\begin{aligned}
T(s'|s,a) &= \frac{\mathbb{E}_{p_u}[\phi(s,a)^T M(u_c)\mu(s') \cdot \pi_\beta(a|s,u_\pi)]}{\mathbb{E}_{p_u}\pi_\beta(a|s,u_\pi)} \\
&= \phi(s,a)^T \left[ \frac{\mathbb{E}_{p_u}[M(u_c)\pi_\beta(a|s,u_\pi)]}{\mathbb{E}_{p_u}\pi_\beta(a|s,u_\pi)} \right] \mu(s') \triangleq \phi(s,a)^T \overline{M}(u)\mu(s').
\end{aligned}
\quad (7)
$$

The derivation of Equation (7) is illustrated in Appendix A.5. Equation (7) basically shows a re-weighting process given the empirical estimation of $M(u_c)$ in every batch of trajectories: $\overline{M}(u) = \frac{\mathbb{E}_{p_u}[M(u_c)\pi_\beta(a|s,u_\pi)]}{\mathbb{E}_{p_u}\pi_\beta(a|s,u_\pi)}$. Compared to the general reweighting strategies in previous MBRL literatures [18, 29] which reweights the entire value function, this re-weighting process is conducted only on the estimated matrix, while the representation $\phi(s,a)$ and $\mu(s')$ are subsequently regularized by weighted estimation of $\overline{M}$. The pipeline of causal world model learning is described in the first part of the Algorithm 1. We discuss more details of the implementation and experiment in Appendix C.

### 3.3 Causal Representation for Uncertainty Quantification

To avoid entering OOD states in the online deployment, we further design a pessimistic planner according to the uncertainty of the predicted trajectories in the imagination rollout step to mitigate objective mismatch.

We use the feature embedding from bilinear causal representation to help quantify the uncertainty, denoted as $E_\theta(s,a)$. As we have access to the offline dataset, we learn an Energy-based Model (EBM) [30, 31] based on the abstracted state representation $\phi$ and core matrix $M$. A higher output of the energy function $E_\theta(\cdot,\cdot)$ indicates a higher uncertainty in the current state as they are visited by the behavior policies $\pi_\beta$ less frequently. In practice, the energy-based model usually suffers from a high-dimensional data space [32]. To mitigate this overhead of training a good uncertainty

---

**Algorithm 1:** BECAUSE Training and Planning

**Input:** Offline dataset $\mathcal{D}$, causal discovery frequency $k$

**Output:** Causal mask $\overline{M}$, feature function $\widehat{\phi}, \widehat{\mu}$, policy $\widehat{\pi}$

```
// Causal world model learning
```
$M_0 \leftarrow [1]^{d' \times d}$

**for** $i \in [K]$ **do**

    Update $\widehat{\phi}_n, \widehat{\mu}_n$ by $\mathcal{L}_{\text{rep}}(\phi,\mu)$ in (6)

    **if** $i \mod k = 0$ **then**

        Update $\widehat{M}_n$ with $\mathcal{L}_{\text{mask}}(M)$ in (5)

        Weighted average $\overline{M}_n$ with (7)

```
// Uncertainty quantifier learning
```
Fit $E_\theta(s,a)$ with $\mathcal{L}_{\text{EBM}}$ (8)

Initialize $\widehat{V}_{H+1}(s) = 0, \forall(s,a)$

```
// Pessimistic planning
```
**while** $h < H$ **do**

    Estimate the uncertainty with score $E_\theta(s,a)$

    Compute $\overline{Q}_h(s,a)$ with (9)

    $\widehat{V}_h(s,a) = \max_a \overline{Q}(s,a)$

    $a \leftarrow \arg\max_a \overline{Q}(s,a)$

    $s', r \leftarrow$ env.step $(a, g)$

---

quantifier, we first embed the state
samples through the abstract representation $\mu(s')$, and the state action pair via $\phi(s, a)$.

$$\mathcal{L}_{\text{EBM}}(\theta) = \mathbb{E}_{\widehat{T}(\cdot|s,a)} E_\theta[\mu(\boldsymbol{s}^+)|\phi(s,a)] - \mathbb{E}_{q(s,a)} E_\theta[\mu(\boldsymbol{s}^-)|\phi(s,a)] + \lambda_{\text{EBM}}\|\theta\|_2, \quad (8)$$

where $\mu(\boldsymbol{s})^+$ refers to the positive samples from the approximated transition dynamics $\widehat{T}(\cdot|s,a)$, and $\mu(\boldsymbol{s}^-)$ refers to the latent negative samples via the Langevin dynamics [30]. Additionally, we regularize the parameters of EBM to avoid overfitting issues. We attach more training details and results of EBMs in Appendix C.2 The learned energy function $E_\theta(s, a)$ is used to quantify the uncertainty based on the offline data.

During the online planning stage, we use the learned EBM to adjust the reward estimation based on Model Predictive Control (MPC) [33]. At timestep $h$, we basically subtract the original step return estimation $r_h(s, a)$ by its uncertainty $E_\theta(s, a)$:

$$\overline{Q}_h(s, a) = \widehat{Q}_h(s, a) - E_\theta(s, a) = \underbrace{r_h(s, a) - E_\theta(s, a)}_{\text{Adjusted Return}} + \sum_{s'\in\mathcal{S}} \widehat{T}(s'|s,a)\widehat{V}_{h+1}(s'). \quad (9)$$

### 3.4 Theoretical Analysis of BECAUSE

Then we move on to develop the theoretical analysis for the proposed method BECAUSE. Based on two standard Assumption 2 and 3 on the feature's existence and regularity, we achieve the finite-sample complexity guarantee — an upper bound of the suboptimality gap as follows, whose proof is postponed to Appendix B.

**Theorem 1** (Performance guarantee). *Consider any $0 < \delta < 1$ and any initial state $\widetilde{s} \in \mathcal{S}$. Under the Assumption 2, 3 and that the transition model $T$ is an SCM (defined in 4), for any accuracy level $0 \le \xi \le 1$, with probability at least $1 - \delta$, the output policy $\pi$ of BECAUSE (Algorithm 1) based on the historical dataset $\mathcal{D}$ with $n = \sum_{(s,a)\in\mathcal{S}\times\mathcal{A}} n(s, a)$ samples generated from a behavior policy $\pi_\beta$ satisfies:*

$$V_1^*(\widetilde{s}) - V_1^\pi(\widetilde{s}) \lesssim \min\left\{C_1 \log\left(\frac{\|M\|_0}{\xi}\right)\sqrt{|\mathcal{S}|}, C_s\sigma\sqrt{\|M\|_0}\right\} \sum_{h=1}^H \mathbb{E}_{\pi^*}\left[\sqrt{\frac{\log(1/\delta)}{n(s_h, a_h)}} \mid s_1 = \widetilde{s}\right],$$

*where $C_1, C_s$ are some universal constants, $\sigma$ is SCM's noise level (see Definition 4), and $M \in \mathbb{R}^{d\times d'}$ is the optimal ground truth sparse transition matrix to be estimated.*

The error bound shrinks as the offline sample size $n$ over all state-action pairs increase. It also grows proportionally to the planning horizon $H$, SCM's noise level $\sigma$, and the $\ell_0$ norm of the ground true causal mask $M$, which describes the intrinsic complexity of the world model.

Consequently, with Proposition 1 in the Appendix, we can achieve $\xi$-optimal policy $(V_1^*(\widetilde{s}) - V_1^\pi(\widetilde{s}) \le \xi)$ as long as the historical dataset satisfies the following conditions: $\forall\, 0 \le \xi \le 1$,

$$\min_{(s,a,h)\in\mathcal{S}\times\mathcal{A}\times[H]} \mathbb{E}_{\pi^\star}\left[n(s_h, a_h) \mid s_1 = \widetilde{s}\right] \gtrsim \frac{\min\left\{C_1^2 \log^2\left(\frac{\|M\|_0}{\xi}\right)|\mathcal{S}|, C_s^2\sigma^2\|M\|_0\right\} \cdot H^2\log(1/\delta)}{\xi^2}.$$

## 4 Experiment Results

In this section, we conduct a comprehensive empirical evaluation of BECAUSE's generalization performance in a diverse set of environments, covering different decision-making problems in the grid world, manipulation, and autonomous driving domains, shown in Figure 4.

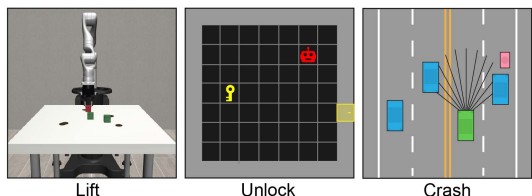

Figure 4: Three environments used in this paper.

### 4.1 Experiment Setting

**Environment Design**  We design 18 tasks in 3 representative RL environments in Figure 4. Agents need to acquire reasoning capabilities to receive higher rewards and achieve goals.

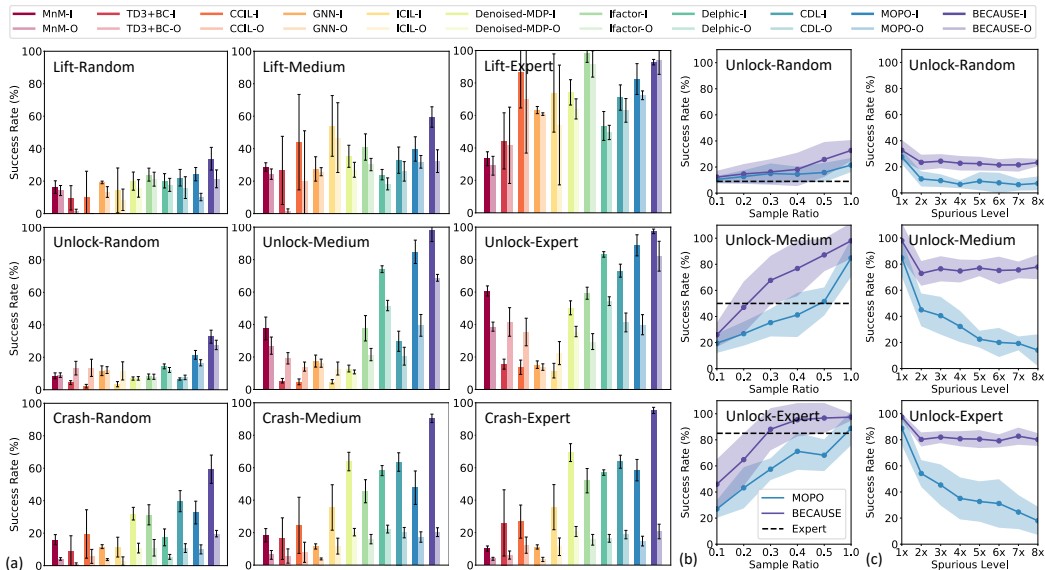

Figure 5: Results of BECAUSE and baselines in different tasks. (a) Average success rate in distribution and out of distribution. (b) Average success rate w.r.t. ratio of offline samples. (c) Average success rate w.r.t. spurious level in the environments. We evaluate the mean and standard deviation of the best performance among 10 random seeds and report task-wise results in Appendix Table 6.

- **Lift**: Object manipulation environment in RoboSuite [34]. We designed this environment for the agent to lift an object with a specific color configuration on the table to a desired height. In the OOD environment *Lift-O*, there is an injected spurious correlation between the color of the cube and the position of the cube in the training phase. During the testing phase, the correlation between color and position is different from training.

- **Unlock**: We designed this environment for the agent to collect a key to open doors in Minigrid [35]. In the OOD environment *Unlock-O*, there will be a different number of goals (doors to be opened) in the testing environments from the training environments.

- **Crash**: Safety is critical in autonomous driving, which is reflected by the collision avoidance capability. We consider a risky scenario where an AV collides with a jaywalker because its view is blocked by another car [36]. We design such a crash scenario based on highway-env [37], where the goal is to create crashes between a pedestrian and AVs. In the OOD environment *Crash-O*, the distribution of reward (number of pedestrians) is different in online testing environments.

For all three different environments, we set a specific subset of the state space as the goal $g \in \mathcal{S}$, and the reward is defined as the goal-reaching reward $r(s, a, g) = \mathbb{I}(r = g)$. When the episode ends in the goal state within the task horizon $H$, the episode is considered a success. We then use the average *success rate* as the general evaluation metrics for our BECAUSE and all baselines.

In each environment, we collect three types of offline data: *random*, *medium*, and *expert* based on the different levels of $u_\pi$ in the behavior policies. In *Unlock* environments, we collect 200 episodes from each level of behavior policies as the offline demonstration data, and the number of episodes is 1,000 in the environments *Lift* and *Crash*, which all have continuous state and action space. A more detailed view of the environment hyperparameters and behavior policy design is in Appendix C.5.

**Baselines**   We compare our proposed BECAUSE with several offline causal RL or MBRL baselines. **ICIL** [38] learns a dynamic-aware invariant causal representation learning to assist a generalizable policy learning from offline datasets. **CCIL** [39] conducts a soft intervention in our offline setting by jointly optimizing policy parameters and masks over the state. **MnM** [9] unifies the objective of jointly training the model and policy, which allocates larger weights in the state prediction loss in the high-reward region. **Delphic** [40] introduces delphic uncertainty to differentiate between uncertainties caused by hidden confounders and traditional epistemic and aleatoric uncertainties. **TD3+BC** [41] is an offline model-free RL approach that combines the Twin Delayed Deep Deterministic Policy

Gradient (TD3) algorithm with Behavior Cloning (BC) to adopt both the actor-critic framework and supervised learning from expert demonstrations. **MOPO** [2] is an offline MBRL approach that uses flat latent space and count-based uncertainty quantification to maintain conservatism in online deployment. **GNN** [42] is a GNN-based baseline using a Relational Graph Convolutional Network to model the temporal dependency of state-action pairs in the dynamic model with message passing. **CDL** [24] uses causal discovery to learn a task-independent world model. **Denoised MDP** [12] and **IFactor** [13] conduct causal state abstraction based on their controllability and reward relevance. The last three methods are designed for online settings, so we only implement their model learning objectives. We attach more details of the baseline implementation in Appendix C.6.

## 4.2 Experiment Results Analysis

We empirically answer the following research questions.

- **RQ1**: How is the generalizability of BECAUSE in the online environments (which may be *unseen*)? Specifically, how does BECAUSE perform under diverse qualities of demonstration data (different level of $u_\pi$), and different environment contexts (different $u_c$)?

- **RQ2**: How does the design in BECAUSE contribute to the robustness of its final performance under different sample sizes or spurious levels?

- **RQ3**: How does BECAUSE scale up to visual RL tasks with image observation input compared to other visual RL baselines?

- **RQ4**: How does BECAUSE achieve the aforementioned generalizability by mitigating the objective mismatch problem in offline MBRL?

For **RQ1**, in Figure 5(a), we evaluate the success rate in the online environment against different baselines. The result shows that under different environments and different qualities of behavior policies $\pi_\beta$ (different $u_\pi$), BECAUSE consistently achieves the best performance in 8 out of 9 for both the in-distribution (I) and out-of-distribution (O) for all the demonstration data quality (different level of $u_\pi$). Where O here indicates the tasks under *unseen* environment with confounder $u_c' \neq u_c$ different from offline training. Another finding is that model-based approaches generally perform better than model-free approaches at various levels of offline data, which shows the importance of world model learning for generalizable offline RL. We attach the detailed results in the Appendix Table 6, 7 and the causal masks discovered in each environment in Appendix Figure 8 for reference.

For **RQ2**, we compare different aspects of BECAUSE's robustness with MOPO without causal structures [2]. We compare their performance with different ratios of the entire offline dataset and illustrate the success rates in Figure 5(b). The result shows that, for any selected number of samples, BECAUSE consistently outperforms MOPO with a clear margin. We also evaluate BECAUSE performance at higher spurious levels in Figure 5(c). We add up to $8\times$ of the original number of confounders in the environments to test the robustness of the agent's performance. BECAUSE consistently outperforms MOPO and the margin enlarges as the spurious level grows higher.

For **RQ3**, we conduct experiments with visual inputs in the Unlock environments with ICIL [38] and IFactor [13]. We parameterize the feature encoder as a three-layer CNN with 128 dimensions hidden size for all the baselines and our methods. The results in Table 1 show that BECAUSE can significantly improve both in-distribution and out-of-distribution performance under different quality of behavior policies.

Table 1: Comparison of visual RL performance.

| Tasks | ICIL | IFactor | BECAUSE |
|---|---|---|---|
| Unlock-I-random | $0.8_{\pm0.8}$ | $4.3_{\pm1.1}$ | $\mathbf{15.7}_{\pm\mathbf{3.3}}$ |
| Unlock-O-random | $1.5_{\pm1.8}$ | $\mathbf{4.7}_{\pm\mathbf{1.6}}$ | $5.9_{\pm0.9}$ |
| Unlock-I-medium | $5.3_{\pm2.0}$ | $30.2_{\pm4.1}$ | $\mathbf{62.0}_{\pm\mathbf{4.6}}$ |
| Unlock-O-medium | $8.6_{\pm4.2}$ | $15.4_{\pm2.4}$ | $\mathbf{71.6}_{\pm\mathbf{9.1}}$ |
| Unlock-I-expert | $8.7_{\pm3.4}$ | $34.0_{\pm4.8}$ | $\mathbf{63.7}_{\pm\mathbf{3.9}}$ |
| Unlock-O-expert | $17.1_{\pm4.2}$ | $16.7_{\pm3.1}$ | $\mathbf{73.6}_{\pm\mathbf{19.5}}$ |

For **RQ4**, we aim to understand whether BECAUSE achieves higher performance by resolving the objective mismatch problem. We first collect two groups of trajectories: $\tau_{pos}$ and $\tau_{neg}$, each with positive reward (success) and negative reward (failure) in *Unlock* task with sparse goal-reaching reward. We want to have a model whose loss is informative for discriminating control results, that is, we wish $\mathcal{L}_{model}(\tau_{pos}) < \mathcal{L}_{model}(\tau_{neg})$. According to our visualization in Figure 6, in *Unlock-Expert* and *Unlock-Medium*, the ratio of $\tau_{pos}$ is much higher in BECAUSE than MOPO among the trajectories with low model loss. In *Unlock-Random*, the mismatch of the model and

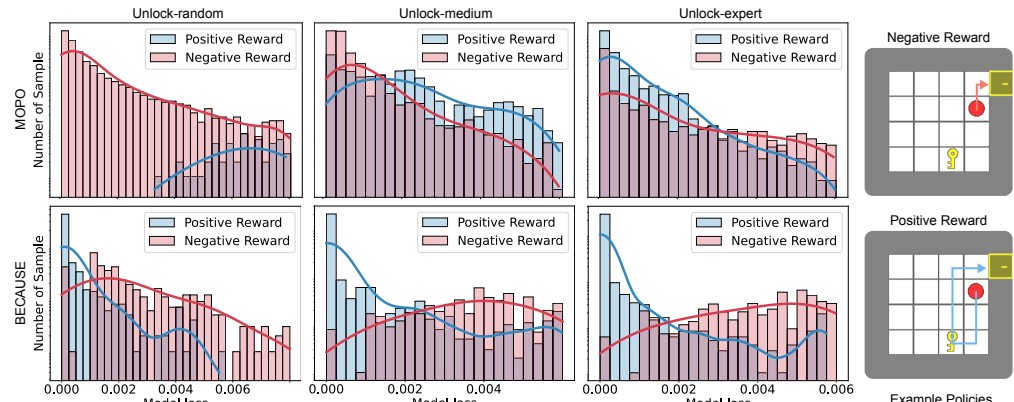

Figure 6: Evaluation of the difference between the distribution of episodic model loss for success and failure trajectories. The higher difference indicates a reduction in model mismatch issues. An example of failure mode is trying to open the door without having the key.

control objective is more significant, since the demonstration is poor in state coverage. MOPO cannot succeed even when the model loss is low, whereas our methods can. We perform a hypothesis test with $H_0 : \mathcal{L}_{model}(\tau_{pos}) < \mathcal{L}_{model}(\tau_{neg})$. In BECAUSE, this desired property is more significant than MOPO attributed to the causal representation we learn, indicating a reduction of objective mismatch. We attach detailed discussions for the mismatch evaluation in Appendix C.3 and Table 5.

## 4.3 Ablation Studies

We conducted ablation studies with three variants of BECAUSE and report the average success rate across nine in-distribution and nine out-of-distribution tasks in Table 2. The **Optimism** variant conducts optimistic planning instead of pessimistic planning in Equation (9), which uses uniform sampling in the planner module. The **Linear** variant assumes a full connection to the causal matrix $M$, then directly uses

Table 2: The ablation studies between BECAUSE and its variants. We report the overall Success rate (%) over 9 in-distribution (I) and 9 out-of-distribution (O) tasks, respectively. **Bold** is the best.

| Variants | BECAUSE | Optimism | Linear | Full |
|---|---|---|---|---|
| Overall-I | **73.3**±**4.5** | 64.4±6.4 | 57.9±6.1 | 39.3±6.3 |
| Overall-O | **43.0**±**4.9** | 32.4±3.3 | 33.2±5.2 | 25.2±3.9 |

linear MDP to parameterize the dynamics model $T$, which removes the causal discovery module in BECAUSE. The **Full** variant learns from the full batch of data to estimate the causal mask without iterative update. We report the results of the task-wise ablation with confidence interval and significance in Appendix Table 8 and 9.

## 5 Related Works

**Objective Mismatch in MBRL** The objective mismatch in MBRL [43, 44] refers to the fact that pure MLE estimation of the world model does not align well with the control objective. Previous works [29, 45] propose reweighting during model training to alleviate this mismatch, [46] proposes a goal-aware prediction by redistributing model error according to their task relevance. These works essentially reweight loss for the entire model training, while our work conducts reweighting just over the estimated causal mask more efficiently. More recently, [9, 10] proposed a joint training between the world model and policies. Although joint optimization improves performance, they do not address the generalizability of the learned model under the distribution shift setting. In the offline setting, Model-based RL [2, 3, 4, 47] employs model ensemble, pessimistic policy optimization or value iteration [48, 49], and an energy-based model for planning [50] to quantify uncertainty and improve test performance. To the best of our knowledge, no previous work explored or modeled the impact of distribution shift on the objective mismatch problem in MBRL.

**Causal Discovery with Confounder** Most of the existing causal discovery methods [51] can be categorized into constraint-based and score-based. Constraint-based methods [52] start from a complete graph and iteratively remove edges with statistical hypothesis testing [53, 54]. This type of method is highly data-efficient but not robust to noisy data. As a remedy, score-based methods [55, 56] use metrics such as the likelihood or BIC [57] as scores to manipulate edges in the causal graph. Recently, researchers have extended score-based methods with RL [58], order learning [59] or differentiable discovery [22, 60, 61]. To alleviate the non-identifiability under hidden confounders, active intervention methods have been explored [62], aiming to break spurious correlations in an online fashion. With extra assumptions on confounders, some recent works detect such correlations [63, 64, 65] so that models can effectively identify elusive confounders.

**Causal Reinforcement Learning** Recently, many RL algorithms have incorporated causality to improve reasoning capability [66] and generalizability. For instance, [67] and [68] explicitly estimate causal structures with the interventional data obtained from the environment in an online setting. These structures can be used to constrain the output space [19] or to adjust the buffer priority [69]. Building dynamic models in model-based RL [24, 70, 71] based on causal graphs is widely studied. Most existing causal MBRL works focus on estimating the causal world model by predicting transition dynamics and rewards. Existing methods learn this causal world model via structural regularization [23, 72, 73], conditional independence test [24, 70, 74], variational inference [75, 12], counterfactual data augmentation [76, 77], hierarchical skill abstraction [78, 79], uncertainty quantification [40], reward redistribution [24, 80], causal context modeling [81, 82] and structure-aware state abstraction [12, 13, 83, 84] based on the controllability and task or reward relevance. However, the presence of confounders during data collection can skew the learned policy, making it susceptible to spurious correlations. Deconfounding solutions have been proposed either between actions and states [39, 85, 86] or among different dimensions of state variables [19, 87].

# 6 Conclusion

In this paper, we study how to mitigate the objective mismatch problem in MBRL, especially under the offline settings where distribution shift occurs. We first propose ASC-MDP and the bilinear causal representation associated with it. Based on the formulation, we proposed how to learn this causal abstraction by alternating between causal mask learning and feature learning in fitting the world dynamics. In the planning stage, we applied the learned causal representation to an uncertainty quantification module based on EBM, which improves the robustness under uncertainty in the online planning stage. We theoretically justify BECAUSE's sub-optimality bound induced by the sparse matrix estimation problem and offline RL. Comprehensive experiments on 18 different tasks show that given a diverse level of demonstration as the offline dataset, BECAUSE has better generalizability than baselines in different online environments, and it robustly outperforms baselines under different spurious levels or sample sizes. We empirically show that BECAUSE mitigates the objective mismatch with causal awareness learned from offline data. One limitation of BECAUSE lies in its simplified assumption of time-homogeneous causal structure, which may not always hold in long-horizon or non-stationary settings. Besides, the current implementation is still based on vector observations. It will be interesting to scale up the causal reasoning framework into high-dimensional observations to discover concept factors in long-horizon visual RL settings.

## Acknowledgement

This work is partially supported by the Defense Advanced Research Projects Agency (DARPA) under Contract No. HR00112320012. The work of L. Shi is supported in part by the Resnick Institute and Computing, Data, and Society Postdoctoral Fellowship at California Institute of Technology.

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

# A  Auxiliary Details of BECAUSE Framework

## A.1  Notation Summary

We illustrate all the notations used in the main paper and appendix in Table 3.

Table 3: Notations used in this paper and their corresponding meanings.

| Notation | Explanation |
|---|---|
| $\mathcal{A}, a$ | Action space, action |
| $\mathcal{S}, s$ | State space, state |
| $r$ | Reward |
| $\gamma$ | Discount factor |
| $T(\cdot|\cdot, \cdot)$ | Transition dynamics |
| $h, H$ | Timestep $h$, Horizon $H$ |
| $\pi^*$ | Optimal policy in the online environments |
| $\pi_\beta$ | Behavior policy generating the offline datasets |
| $V(\cdot)$ | State value function |
| $Q(\cdot, \cdot)$ | Action-state value function |
| $\mathcal{D}$ | Offline datasets |
| $n(s, a)$ | Number of samples for (s, a) pairs in offline datasets |
| $\mathbb{B}_h$ | Bellman operator |
| $\phi(\cdot, \cdot)$ | Feature representation of state and action |
| $\widetilde{\phi}(\cdot, \cdot)$ | Feature of state, action and confounder in Equation (2) |
| $\mu(\cdot)$ | Feature representation of next state |
| $d, d'$ | Dimensions of features for $\phi(\cdot, \cdot)$ and $\mu(\cdot)$ |
| $K_\mu$ | Feature matrix expanded from $\mu$ in Equation (5) |
| $C_\phi, C_\mu, C_\beta$ | Feature regularity |
| $C_s$ | Sparsity-related constant |
| $\kappa$ | Restrictive eigenvalue constant |
| $X$ | Feature Kronecker product |
| $u, u_c, u_\pi$ | Confounders |
| $M, M(u)$ | Binary transition matrix (under certain confounders) |
| $G$ | Causal graph |
| $\widehat{M}, \overline{M}(u)$ | Estimated causal matrix |
| $\beta^M, \widehat{\beta}^M$ | Optimal / estimated parameters in causal matrix |
| $\mathbf{PA}^G(\cdot)$ | Parental node in the causal graph $G$ |
| $\epsilon$ | Exogenous noise in SCM by Definition 4 |
| $\lambda, \lambda_\phi, \lambda_\mu$ | Spectrum regularizer weight in Equation (6) |
| $\sigma$ | Standard deviation of exogenous noise in SCM |
| $\xi$ | Accuracy level of the policy |
| $K$ | Iterative update steps in BECAUSE |
| $E_\theta$ | Energy-based model |
| $\delta$ | Level of 'high probability' |
| $\Gamma(\cdot, \cdot)$ | Uncertainty quantification function |
| $\mathcal{E}$ | $\delta$-uncertainty quantifier set |
| $\lambda_{\text{EBM}}$ | Regularizer weight for the $\ell_2$ norm in EBM |

## A.2  Equivalence of the Assumptions with Previous Works

## A.3  Derivation of Definition 3

The node of this causal graph $G = \begin{bmatrix} 0^{d \times d} & M \\ 0^{d \times d'} & 0^{d \times d} \end{bmatrix}$ contains two groups of entities: (1) The state action abstraction $\phi(s, a)$, and (ii) the next state abstraction $\mu(s')$. We denote $\phi(\cdot, \cdot)^{(i)}$ as the $i^{th}$

Table 4: Summary of assumptions and equivalent forms.

| Assumption | Base and Equivalent Format |
|---|---|
| Invariant State Abstraction [25] | $T(s'\|s,a;u) = T(\phi(s')\|\phi(s),a) \cdot p_e(u'\|s,u)$ 
 $T(s'\|s,a;u) \propto T(\phi(s')\|\phi(s),a)$ |
| Task Independence [24] | $T(\phi(s')\|\phi(s),a) = T(s'_C\|s_C,s_R,a)p(s'_R\|s_R)$ 
 $T(\phi(s^{(1)})\|\phi(s^{(1)}),a;u^{(1)}) = T(\phi(s^{(2)})\|\phi(s^{(2)}),a;u^{(2)})$ |
| Invariant Action Effect [26] | $T_{\text{online}}(s'\|s,a) = \sum_u T(s'\|s,a,u)\hat{p}(u\|s)$ 
 $T_{\text{offline}}(s'\|s,a) = \sum_u T(s'\|s,a,u)\hat{p}(u\|s)$ 
 $T_{\text{online}}(s'\|s,a) \propto T_{\text{offline}}(s'\|s,a)$ |
| Invariant Causal Graph | $u^{(1)} \neq u^{(2)}, M(\mathbf{u}^{(1)}) = M(u^{(2)})$ 
 $T(s'\|s,a;u^{(1)}) = T(s'\|s,a;u^{(2)})$ |

factor in the abstracted state action representations, and $\mu(\cdot)^{(j)}$ for the $j^{th}$ factor in the abstracted state representations.

The source node of all the nodes $G$ is $\phi(s,a)^{(i)}$, which is the abstracted state-action representation, and the sink node of all the edges in $G$ is the to $\mu(s)'^{(i)}$.

$$T(s'|s,a) = \begin{bmatrix} \phi(s,a)^T & \mu(s')^T \end{bmatrix} \begin{bmatrix} 0^{d\times d} & M \\ 0^{d\times d'} & 0^{d\times d} \end{bmatrix} \begin{bmatrix} \phi(s,a) \\ \mu(s') \end{bmatrix} = \phi(s,a)^T M \mu(s'), \qquad (10)$$

Therefore, $G$ is a bipartite graph, since there will be no edges between $\phi(s,a)^{(i)}, \phi(s,a)^{(j)}$, or $\mu(s)^{(i)}, \mu(s)^{(j)}$.

Consequently, we show that $G \in \text{DAG}$.

### A.4 Derivation of Equation (5)

**Definition 4** (Structured Causal Model). An SCM $\theta := (\mathcal{S}, \mathcal{E})$ consists of a collection $\mathcal{S}$ of $d$ functions [11],

$$s_j := f_j(\mathbf{PA}^G(s_j), \epsilon_j), \ \ j \in [d], \qquad (11)$$

where $\mathbf{PA}_j^G \subset \{s_1, \ldots, s_d\} \backslash \{s_j\}$ are called parents of $x_j$ in the Directed Acyclic Graph (DAG) $G$, and $\mathcal{E} = \{\epsilon_i\}_{i=1}^d$ are jointly independent. For instance, in continuous state and action space, we parameterize the world model with joint Gaussian Distribution, i.e. $\epsilon \sim \mathcal{N}(0, \sigma I_{dd'})$.

We then use bilinear MDP to approximate the original likelihood function in Equation (4), i.e.

$$p(\mathcal{D}; \phi, \mu, M) \propto \prod_{(s,a,s') \in \mathcal{D}} \exp(-\|\mu^T(s')K_\mu^{-1} - \phi^T(s,a)M\|_2^2), \qquad (12)$$

where $K_\mu := \sum_{s' \in \mathcal{S}} \mu(s')\mu(s')^T$ is an invertible matrix. Then we can apply an MLE in Equation (5).

In our BECAUSE algorithm, the optimization of the causal world model is conducted by solving the regularized MLE problem in Equation (13). The biggest difference between BECAUSE and the offline version of [14, 15] is that it aims to apply $\ell_0$ regression instead of ridge regression to estimate matrix $M$:

$$M_n = \arg\max_M \left[ \log p(\mathcal{D}; \phi, \mu, M) - \lambda|M| \right]$$

$$= \arg\min_M \underbrace{\sum_{(s,a,s') \in \mathcal{D}} \|\mu^T(s')K_\mu^{-1} - \phi^T(s,a)M\|_2^2}_{\text{World Model Learning}} + \underbrace{\lambda\|M\|_0}_{\text{Sparsity Regularization}}. \qquad (13)$$

## A.5 Proof of Equation (7)

The derivation depends on the following re-weighting formula in [18]:

$$T(s'|s,a) = \frac{\mathbb{E}_{p_u} T(s'|s,a,u)\pi_\beta(a|s,u_\pi)}{\mathbb{E}_{p_u} \pi_\beta(a|s,u_\pi)}. \tag{14}$$

Then we apply equation (14) to the decomposition in Equation (2) and Equation (3), which yields

$$
\begin{aligned}
T(s'|s,a) &= \frac{\mathbb{E}_{p_u}\big[T(s'|s,a,u)\pi_\beta(a|s,u_\pi)\big]}{\mathbb{E}_{p_u} \pi_\beta(a|s,u_\pi)} \\
&= \frac{\mathbb{E}_{p_u}\big[\phi(s,a)^T M(u_c)\mu(s')\pi_\beta(a|s,u_\pi)\big]}{\mathbb{E}_{p_u}[\pi_\beta(a|s,u_\pi)]} \\
&= \phi(s,a)^T \left[\frac{\mathbb{E}_{p_u}\big[M(u_c)\pi_\beta(a|s,u_\pi)\big]}{\mathbb{E}_{p_u}[\pi_\beta(a|s,u_\pi)]}\right]\mu(s') \\
&= \phi(s,a)^T \overline{M}(u)\mu(s'),
\end{aligned}
\tag{15}
$$

where the last equality holds by letting $\overline{M}(u) := \frac{\mathbb{E}_{p_u}[M(u_c)\pi_\beta(a|s,u_\pi)]}{\mathbb{E}_{p_u}[\pi_\beta(a|s,u_\pi)]}$.

# B Proof of Theorem 1

In this section, we provide the proof of the sub-optimality upper bound in Theorem 1. We first show some useful definitions and lemmas in Section B.1. Armed with them, we provide the theoretical results tailored for the causal discovery setting in Section B.2. Furthermore, we give a detailed proof of the uncertainty set form in our causal discovery problems in Section B.3.

## B.1 Preliminary

In this subsection, we first define the $\delta$-uncertainty quantifier $\Gamma$, then we refer to the lemmas in the previous literature to construct a suboptimality bound based on the defined uncertainty quantifier $\Gamma$.

First, we define the Bellman operator $\mathbb{B}_h$, for some value function $V : \mathcal{S} \mapsto \mathbb{R}$, the Bellman operator can be defined as:

$$(\mathbb{B}_h V)(s,a) = \mathbb{E}[r_h(s_h,a_h) + V(s_{h+1})|s_h = s, a_h = a]. \tag{16}$$

Similarly, we denote the approximate Bellman operator of the empirical MDP constructed from the offline dataset $\mathcal{D}$ as $\widehat{\mathbb{B}}_h$ for any $h \in [H]$.

**Definition 5** ($\delta$-Uncertainty Quantifier). We let $\{\Gamma_h\}_{h=1}^H$, $\Gamma_h : \mathcal{S} \times \mathcal{A} \mapsto \mathbb{R}$ to be a $\delta$-uncertainty quantifier with respect to data distribution $P_\mathcal{D}$ if the event:

$$\mathcal{E} = \Big\{|(\widehat{\mathbb{B}}_h\widehat{V}_{h+1})(s,a) - (\mathbb{B}_h\widehat{V}_{h+1})(s,a)| \leq \Gamma_h(s,a), \forall(s,a,h) \in \mathcal{S} \times \mathcal{A} \times [H]\Big\}$$

satisfies $\mathbb{P}_\mathcal{D}(\mathcal{E}) \geq 1 - \delta$.

As we consider the offline model learning and planning, we define the model evaluation error at each step $h \in [H]$ as

$$\forall(s,a) \in \mathcal{S} \times \mathcal{A}: \quad \iota_h(s,a) = (\mathbb{B}_h\widehat{V}_{h+1})(s,a) - \widehat{Q}_h(s,a), \tag{17}$$

where $\iota_h$ is the error induced by the approximate Bellman operator, especially the transition kernel based on $\mathcal{D}$. We then identify the source of sub-optimality in our offline MBRL setting by decomposing the sub-optimality error in Lemma 1.

**Lemma 1** (Decomposition of Suboptimality [88]).

$$
\begin{aligned}
\forall s \in \mathcal{S}: \quad V_h^*(s) - V_h^\pi(s) &= -\sum_{h'=h}^H \mathbb{E}_\pi[\iota_h(s_{h'},a_{h'})|s_h = s] + \sum_{h'=h}^H \mathbb{E}_{\pi^*}[\iota_{h'}(s_{h'},a_{h'})|s_h = s] \\
&+ \sum_{h'=h}^H \mathbb{E}_{\pi^*}[\langle\widehat{Q}_{h'}(s_{h'},\cdot), \pi^*(\cdot,s_{h'}) - \widehat{\pi}(\cdot,s_{h'})\rangle_\mathcal{A}|s_h = s],
\end{aligned}
\tag{18}
$$

*where $\pi$ is any learned policy, $\pi^*$ is the optimal policy that maximizes the cumulative return as below:*

$$\pi^* = \arg\max_\pi \mathbb{E}_\pi\Big[\sum_{h'=1}^H \gamma^{h'} r(s_{h'}, a_{h'})|s_h\Big].$$

Based on this decomposition, we will get the basic form of sub-optimality error bound for general offline RL settings in Lemma 2:

**Lemma 2** (Suboptimality in standard MDP [88]). *Suppose we have $\{\Gamma_h\}_{h=1}^H$ as $\delta$-uncertainty quantifier. Under $\mathcal{E}$ defined in Equation (5), the suboptimality error bound by conservative planning satisfies:*

$$\forall s \in \mathcal{S}: \quad V_h^*(s) - V_h^\pi(s) \le 2\sum_{h'=h}^H \mathbb{E}_{\pi^*}[\Gamma_{h'}(s_{h'}, a_{h'})|s_1 = s].$$

The basic form of sub-optimality bound in Lemma 2 involves an uncertainty quantifier $\Gamma_h$, which in our case will be further replaced by an exact bound in our sparse matrix estimation problem of causal discovery algorithms.

## B.2   Proof of Theorem 1

The main results hold under the following two assumptions:

**Assumption 2** (Existence of a core matrix given the feature embedding). For each $(s, a) \in \mathcal{S} \times \mathcal{A}$, feature vectors $\phi(s, a) \in \mathbb{R}^d, \mu(s) \in \mathbb{R}^{d'}$ are approximated as a priori. Given a specific confounder set $u$, there exists an unknown matrix $M(u)^* \in \mathbb{R}^{d' \times d}$ such that,

$$T(s'|s, a, u) = \phi(s, a)^T M(u)\mu(s'). \tag{19}$$

**Assumption 3** (Feature regularity). We assume feature regularity [14, 15] for the following components of the confounded bilinear MDP:

- $\forall u, \|M(u)\|_F^2 \le C_M d$,

- $\forall (s, a) \in \mathcal{S} \times \mathcal{A}, \|\phi(s, a)\|_2^2 \le C_\phi d$,

- $\forall s' \in \mathbb{R}^{|\mathcal{S}|}, \|\mu^T s'\|_2 \le C_\mu \|s'\|_\infty, \|\mu K_\mu^{-1}\|_{2,\infty} \le C_\mu'$,

- $\forall s, a, s' \in \mathcal{S} \times \mathcal{A} \times \mathcal{S}, \|\phi(s, a)\mu(s')^T\|_1 \le C_\mu$.

where $C_M, C_\phi, C_\mu, C_\mu'$ are some universal constants.

Here, for any matrix $X$, $\|X\|_{2,\infty} := \max_i \sqrt{\sum_j X_{ij}^2}$ represents the operator $2 \mapsto \infty$ norm.

**Proof pipeline.**   Armed with the above assumptions, we turn to the bilinear MDP setting, which this work focuses on. We shall develop the finite-sample analysis by specifying the main error term —-$\delta$-uncertainty quantifier $\Gamma$ (see Lemma 2) for our time-homogeneous core matrix estimation problem in the following lemma.

**Lemma 3** (Uncertainty bound for Bilinear Causal Representation). *Under the Assumption 2, 3 and that $T$ is an SCM (defined in 4), for the BECAUSE algorithm, for the $\xi$-optimal policy ($V_1^*(\tilde{s}) - V_1^\pi(\tilde{s}) \le \xi$), $\forall 0 \le \xi \le 1$, we have the $\delta$-uncertainty set as:*

$$\mathcal{E}_{BECAUSE} = \Big\{ |(\widehat{\mathbb{B}}_h \widehat{V}_{h+1})(s, a) - (\mathbb{B}_h \widehat{V}_{h+1})(s, a)|$$

$$\lesssim \min\Big\{ C_1 \log\big(\frac{\|M\|_0}{\xi}\big)\sqrt{|\mathcal{S}|}, C_s\sigma\sqrt{\|M\|_0}\Big\}\sqrt{\frac{\log(1/\delta)}{n(s, a)}}, \forall (s, a, h) \in \mathcal{A} \times \mathcal{S} \times [H] \Big\},$$

*where $C_1$ is some universal constants.*

Armed with the above lemma, we complete the proof of Theorem 1 by showing that

$$V_1^*(\widetilde{s}) - V_1^\pi(\widetilde{s}) \leq 2 \sum_{h'=1}^{H} \mathbb{E}_{\pi^*}[\Gamma_{h'}(s_{h'}, a_{h'})|s_1 = \widetilde{s}]$$

$$\lesssim 2 \sum_{h=1}^{H} \mathbb{E}_{\pi^*}\left[\min\left\{C_1 \log\left(\frac{\|M\|_0}{\xi}\right)\sqrt{|\mathcal{S}|}, C_s \sigma \sqrt{\|M\|_0}\right\}\sqrt{\frac{\log(1/\delta)}{n(s_h, a_h)}} \mid s_1 = \widetilde{s}\right] \quad (20)$$

This concludes the proof of Theorem 1.

### B.3 Proof of Lemma 3

The key to proving Theorem 1 is to prove Lemma 3. The proof pipeline of Lemma 3 is illustrated below. In **Step 1**, we derive the estimation of the causal transition matrix $M$ in BECAUSE as a sparsity regression problem. In **Step 2**, we decompose the error terms within $\delta$-uncertainty set into two parts: (a) error due to the under-explored dataset, (b) error due to optimization error in the structured causal model. Then we bound both error terms in **Step 3** and **Step 4**, respectively. Finally, in **Step 5**, we sum up all the results and derive the form of $\delta$-uncertainty quantifier which will lead to our final results in Theorem 1.

**Step 1: deriving the output model of BECAUSE.** Recalling the original optimization problem in equation (13) to estimate the core matrix:

$$\widehat{M} = \arg\max_M [\log p(\phi, \mu, M) - \lambda|M|]$$

$$= \arg\min_M \underbrace{\sum_{(s,a,s')\in\mathcal{D}} \|\mu(s')^T K_\mu^{-1} - \phi(s,a)^T M\|_2^2}_{\text{Model Learning}} + \underbrace{\lambda\|M\|_0}_{\text{Sparsity Regularization}}. \quad (21)$$

This part of derivation aims to transform the above estimation problem into a linear regression problem, with the regression data pairs $(X, T)$ and some unknown parameters $\beta$ associated with mask $M$ to be estimated. Eventually, we'll derive the representation of each part of $\beta^M, X, T$, and eventually reach the following form:

$$\min_{\beta^M} \sum_{(s_i, a_i, s_i')\in\mathcal{D}} [\|T_{\pi_\beta}(s_i' \mid s_i, a_i) - X_i\beta^M\|_2^2 + \lambda\|\beta^M\|_0]. \quad (22)$$

We define each component of this target form of $\ell_0$ regression as follows:

- **For unknown parameters $\beta^M$:** We first define $\beta^M \in [0, 1]^{dd'}$ as a column dimensional vector consisting of all the entries in time-homogenous causal matrix $M$, where $\beta_i^M$ denotes the $i$-th entry of $\beta^M$. Besides, we define $\beta_\mathcal{D}^M$ as the true core matrix given some offline dataset $\mathcal{D}$ and corresponding data pairs $T_{data}, X_{data}$ that satisfies $T_{data} = \beta_\mathcal{D}^M X_{data} + \epsilon$.

- **For dataset $\mathcal{D}$:** Recall the transition pairs in the offline dataset $\mathcal{D} = \{s_i, a_i, s_i'\}_{1 \leq i \leq n}$. Here, $n$ represents the sample size over certain state-action pairs in the rollout data by some behavior policy $\pi_\beta$. For simplicity, we denote $n \triangleq n(s, a)$ in the following derivation, which is mentioned in Section 2.1.

- **For regression target $T_{\pi_\beta}$:** Then, we introduce the following transition targets $T_{\pi_\beta}$ induced by the offline dataset $\mathcal{D}$ sampled with behavior policy $\pi_\beta$:

$$T_{\pi_\beta}(s'|s, a) := \frac{\sum_{(s_i, a_i, s_i')\in\mathcal{D}} \mathbf{1}(s_i = s, a_i = a, s_i' = s')}{\sum_{(s_i, a_i, s_i')\in\mathcal{D}} \mathbf{1}(s_i = s, a_i = a)}$$

$$= \frac{1}{n(s, a)} \sum_{(s_i, a_i, s_i')\in\mathcal{D}} \mathbf{1}(s_i = s, a_i = a, s_i' = s'). \quad (23)$$

Under the $n$ finite samples in the offline dataset, we assume that $T_{\pi_\beta} \sim \mathcal{N}(\mathbb{E}[T_{\pi_\beta}], \sigma^2 I_n)$. The above definition specifies the regression target in the $\ell_0$ regression problem, and we denote

$T_{\pi_\beta} = [T_{\pi_\beta}(s'_1|s_1, a_1), \cdots, T_{\pi_\beta}(s'_n|s_n, a_n)]^T \in \mathbb{R}^n$ as the empirical transition probabilities of certain transition pairs in the offline data $\mathcal{D} = \{s_i, a_i, s'_i\}_{1 \le i \le n}$.

- **For regression data** $X$: Next, we need to specify the data $X$ in the regression problem. We denote the i-th row of $X$ as the i-th sample in the offline transition pairs $X_i \in \mathcal{D}$, which is a vector of Kronecker product between $\phi(s_i, a_i) \in \mathbb{R}^d$ and normalized $\frac{\mu(s'_i)}{C_\mu} \in \mathbb{R}^{d'}$ (without loss of generality, we assume $C_\phi = 1$ and only need to normalize $\mu(s'_i)$ by $C_\mu$):

$$
\begin{aligned}
X_i &= \phi(s_i, a_i) \otimes \frac{\mu(s'_i)}{C_\mu} \\
&= \frac{1}{C_\mu} [\phi(s_i, a_i)^{(1)} \mu(s'_i)^{(1)}, \phi(s_i, a_i)^{(1)} \mu(s'_i)^{(2)}, \cdots, \phi(s_i, a_i)^{(d)} \mu(s'_i)^{(d')}]^T
\end{aligned}
\tag{24}
$$

As a result, $X_i \in \mathbb{R}^{dd'}$, since there are in all $n$ samples in offline dataset, $X \in \mathbb{R}^{n \times dd'}$ is the dataset-dependent matrix with all $n$ rows of samples, and $d$ and $d'$ are the latent dimension of $\phi$ and $\mu$, respectively. Based on the feature regularity criteria in Assumption 3, we have $\|X_i\|_2 \le \|X_i\|_1 \le 1, \|X\|_\infty \le 1$.

The prior work [14] estimate the transition kernel of a bilinear MDP using the following ridge regression:

$$
\min_M \mathbb{E}_{(s,a,s') \in \mathcal{D}} \| \mu(s')^T K_\mu^{-1} - \phi(s, a)^T M \|_2^2 + \lambda \|M\|_2.
\tag{25}
$$

In this paper, in order to promote the sparsity of the matrix $M$, we introduce the $\ell_0$ regularization term and arrive at the following optimization problem:

$$
\begin{aligned}
&\min_{\beta^M} \sum_{(s_i, a_i, s'_i) \in \mathcal{D}} [\|\mu(s'_i)^T K_\mu^{-1} \mu(s'_i) - \phi(s_i, a_i)^T M \mu(s'_i)\|_2^2 + \lambda \|\beta^M\|_0] \\
&\rightarrow \min_{\beta^M} \sum_{(s_i, a_i, s'_i) \in \mathcal{D}} [\|T_{\pi_\beta}(s'_i \mid s_i, a_i) - X_i \beta^M\|_2^2 + \lambda \|\beta^M\|_0] =: \widehat{\beta}_\mathcal{D}^M,
\end{aligned}
\tag{26}
$$

where we denote the solution associated with the offline dataset $\mathcal{D}$ as $\widehat{\beta}_\mathcal{D}^M$. Here, we use the empirical version constructed by the finite samples in offline dataset $\mathcal{D}$.

Given the goal-conditioned reward setting, for a single episode $s \sim \tau$, $r(s, a; g) = 1$ if and only if $s = g$, otherwise $r(s, a; g) = 0$, as is specified in Section 2.1. Since we are essentially predicting the probabilities (normalized to a sum of 1) of whether the next state is the goal state, i.e. $\sum_{s \in \mathcal{S}} \widehat{V}(s) = 1$. Therefore, we have $\|\widehat{V}(\cdot)\|_1 \le 1$.

As is denoted by Equation (23), for the specific offline dataset collected by some behavior policies $\pi_\beta$, we have the regression target $T_{\pi_\beta}(s'|s, a) = \frac{\sum_{(s_i, a_i, s'_i) \in \mathcal{D}} \mathbf{1}(s_i = s, a_i = a, s'_i = s')}{\sum_{(s_i, a_i, s'_i) \in \mathcal{D}} \mathbf{1}(s_i = s, a_i = a)}$, and the corresponding features induced by the dataset $X_i = \phi(s_i, a_i) \otimes \frac{\mu(s'_i)}{C_\mu}$. We have the following equations hold:

$$
T_{\pi_\beta} = X \beta_\mathcal{D}^M + \epsilon,
\tag{27}
$$

where $\beta_\mathcal{D}$ is the true underlying transition mask given the offline dataset $\mathcal{D}$, $\epsilon \sim \mathcal{N}(0, \sigma \cdot I_n)$ is some exogenous noise of the transition model.

Specifically, for the regression problem, we transform the original trajectory dataset $\mathcal{D}$ as $[X, T_{\pi_\beta}]$, as the representation of transition pairs rolled out by the behavior policy $\pi_\beta$. Similarly, we can define some 'well-explored dataset' $\mathcal{D}^*$, which is an infinite dataset rollout by the behavior policy $\pi_\beta$, $\mathcal{D}^* = \{s_i, a_i, r_i\}_{i=0}^\infty$, similar to the definition of regression target $T$ in Equation (23) and representation data $X$ in Equation (24), we have $[X, \mathbb{E}[T_{\pi_\beta}]]$ as the regression pairs with access to the true transition probability distribution for all the state action pairs $(s, a)$. Here $X \in \mathbb{R}^{n \times dd'}$ is the regression data defined in equation (22). Recalling the definition in Equation (23), we can further build the regression target under well-explored dataset $\mathcal{D}^*$ as:

$$
\begin{aligned}
\mathbb{E}[T_{\pi_\beta}(s'|s, a)] &= \frac{\sum_{(s_i, a_i, s'_i) \in \mathcal{D}^*} \mathbf{1}(s_i = s, a_i = a, s'_i = s')}{\sum_{(s_i, a_i, s'_i) \in \mathcal{D}^*} \mathbf{1}(s_i = s, a_i = a)} \\
&= \mathbb{E}_{s' \sim T(\cdot|s, a)} [\mathbf{1}(s_i = s, a_i = a, s'_i = s')]
\end{aligned}
\tag{28}
$$

We denote $\mathbb{E}[T_{\pi_\beta}] = \left[\mathbb{E}[T_{\pi_\beta}(s'_1|s_1,a_1)], \cdots \mathbb{E}[T_{\pi_\beta}(s'_n|s_n,a_n)]\right]^T \in \mathbb{R}^n$. In practice, with finite sample size $n$, we have $\mathbb{E}[T_{\pi_\beta}(s'|s,a)] = T(s'|s,a) + \epsilon = X\beta_{\mathcal{D}^*}^M + \epsilon$. In addition, we also introduce a vector form $\mathbf{T} \in \mathbb{R}^{|\mathcal{S}||\mathcal{A}| \times |\mathcal{S}|}$ so that $\forall(s,a) \in \mathcal{S} \times \mathcal{A}$, we have

$$\mathbf{T}(\cdot|s,a) = \begin{bmatrix} T(s'_1|s,a) & T(s'_2|s,a) & \cdots & T(s'_{|\mathcal{S}|}|s,a) \end{bmatrix}^T \in \mathbb{R}^{|\mathcal{S}|},$$

where the state space is denoted as $\mathcal{S} = \{s'_1, \cdots, s_{|\mathcal{S}|}\}$ are all possible states. Similar to the Kronecker product we define for $X$ in equation (26), we define $\mathbf{X}$ in a matrix form for any state-action pair $(s,a)$:

$$\mathbf{X}(\cdot|s,a) = \left[\phi(s,a) \otimes \frac{\mu(s'_1)}{C_\mu}, \cdots \phi(s,a) \otimes \frac{\mu(s'_{|\mathcal{S}|})}{C_\mu}\right]^T \in \mathbb{R}^{|\mathcal{S}| \times dd'}.$$

In addition, we let $\mathbf{X}(s'|s,a) \in \mathbb{R}^{1 \times dd'}$ denote the $s'$-th row of $\mathbf{X}(\cdot|s,a)$ associated with the state $s' \in \mathcal{S}$. Consequently, the estimated transition kernel can be expressed as follows:

$$\widehat{\mathbf{T}}(\cdot|s,a) = \phi^T(s,a)\widehat{M}\mu(\cdot) = \mathbf{X}(\cdot|s,a)\widehat{\beta}_{\mathcal{D}}^M.$$

**Step 2: decomposing the term of interest.** To begin with, recalling the definition of Bellman operator $\mathbb{B}_h$ in Equation (16) and applying Hölder's inequality, the term of interest for any time step $1 \le h \le H$ and state-action pair $(s,a) \in \mathcal{S} \times \mathcal{A}$ can be controlled as

$$
\begin{aligned}
|(\widehat{\mathbb{B}}_h \widehat{V}_{h+1})(s,a) - (\mathbb{B}_h \widehat{V}_{h+1})(s,a)| &\le |\langle \widehat{\mathbf{T}}(\cdot|s,a) - \mathbf{T}(\cdot|s,a), \widehat{V}_{h+1}\rangle| \\
&\le \|\widehat{\mathbf{T}}(\cdot|s,a) - \mathbf{T}(\cdot|s,a)\|_\infty \|\widehat{V}_{h+1}\|_1 \\
&\le \|\widehat{\mathbf{T}}(\cdot|s,a) - \mathbf{T}(\cdot|s,a)\|_\infty,
\end{aligned}
\tag{29}
$$

where the first inequality is held given our goal-conditioned reward formulation in section 2.1. To continue, we have

$$
\begin{aligned}
|(\widehat{\mathbb{B}}_h \widehat{V}_{h+1})(s,a) - (\mathbb{B}_h \widehat{V}_{h+1})(s,a)| &\le \|\widehat{\mathbf{T}}(\cdot|s,a) - \mathbf{T}(\cdot|s,a)\|_\infty \\
&= \|\mathbf{X}(\cdot|s,a)\widehat{\beta}_{\mathcal{D}}^M - \mathbf{X}(\cdot|s,a)\beta^M\|_\infty
\end{aligned}
\tag{30}
$$

Here, we recall $\widehat{\beta}_{\mathcal{D}}^M$ represents the parameter vector in the estimated causal masks based on the offline dataset $\mathcal{D}$ sampled by $\pi_\beta$. Similarly, we denote $\widehat{\beta}_{\mathcal{D}^*}^M$ as the estimated causal mask outputted from equation (26) based on the infinite dataset $\mathcal{D}^*$ generated by the behavior policy $\pi_\beta$. Then, we can further control equation (30) as

$$
\begin{aligned}
\|\mathbf{X}(\cdot|s,a)\widehat{\beta}_{\mathcal{D}}^M - \mathbf{X}(\cdot|s,a)\beta^M\|_\infty &= \|\mathbf{X}(\cdot|s,a)\widehat{\beta}_{\mathcal{D}}^M - \mathbf{X}(\cdot|s,a)\widehat{\beta}_{\mathcal{D}^*}^M + \mathbf{X}(\cdot|s,a)\widehat{\beta}_{\mathcal{D}^*}^M - \mathbf{X}(\cdot|s,a)\beta^M\|_\infty \\
&\le \|\mathbf{X}(\cdot|s,a)[\widehat{\beta}_{\mathcal{D}}^M - \widehat{\beta}_{\mathcal{D}^*}^M]\|_\infty + \|\mathbf{X}(\cdot|s,a)[\widehat{\beta}_{\mathcal{D}^*}^M - \beta^M]\|_\infty \\
&\le \|\mathbf{X}(\cdot|s,a)\|_\infty \|\widehat{\beta}_{\mathcal{D}}^M - \widehat{\beta}_{\mathcal{D}^*}^M\|_\infty + \|\mathbf{X}(\cdot|s,a)\|_\infty \|\widehat{\beta}_{\mathcal{D}^*}^M - \beta^M\|_\infty \\
&\le \underbrace{\|\widehat{\beta}_{\mathcal{D}}^M - \widehat{\beta}_{\mathcal{D}^*}^M\|_\infty}_{(a)} + \underbrace{\|\widehat{\beta}_{\mathcal{D}^*}^M - \beta^M\|_\infty}_{(b)}
\end{aligned}
\tag{31}
$$

Here the last inequality comes from the fact that

$$
\begin{aligned}
\|\mathbf{X}(\cdot|s,a)\|_\infty &= \max_{i \in |\mathcal{S}|} \sum_{j \in [dd']} |\mathbf{X}(\cdot|s,a)_{ij}| = \max_{i \in |\mathcal{S}|} \|\mathbf{X}(\cdot|s,a)_i\|_1 \\
&= \max_{i \in |\mathcal{S}|} \|\phi(s,a)\mu(s'_i)^T\|_1 \le \frac{C_\mu}{C_\mu} = 1
\end{aligned}
$$

based on the definition of $X$ in equation (24) and assumption 3. Here $(a)$ comes from the mismatch error between the demonstrated offline dataset and some optimal rollout datasets. And $(b)$ comes from the error of the $\ell_0$ optimization of causal masks given the existence of exogenous noise $\sigma$ defined by SCM in Definition 4. We will control them separately in the following.

**Step 3: Controlling term (a).** We need to consider the optimization process in the original regression problem in equation (22) to fully understand the difference between $\widehat{\beta}_{\mathcal{D}}^M$ and $\widehat{\beta}_{\mathcal{D}^*}^M$, where the only difference is that the latter uses a perfect dataset with infinite samples. The optimization problem we target (cf. equation (26)) can be solved by the iterative hard thresholding algorithm (IHT) proposed by [89] IHT offers an iterative solution for the $\ell_0$ regression problem, armed with a hard thresholding operator as below:

$$[g_\lambda(\beta)]_j = \begin{cases} \max\{0, \beta_j - \lambda\} & \text{if } \beta_j > \lambda \\ 0 & \text{if } \beta_j \leq \lambda, \ j = 1, \cdots, dd'. \end{cases} \tag{32}$$

We denote $\widehat{\beta}_{\mathcal{D}}^M(i)$ as the estimated causal mask parameters after $i$-th iterations with dataset $\mathcal{D}$. Similarly, we denote $\widehat{\beta}_{\mathcal{D}^*}^M(i)$ as the estimation after $i$-th iterations with dataset $\mathcal{D}^*$. We initialize the graph to be a full graph regardless of the datasets ($\mathcal{D}^*$ or $\mathcal{D}$) used in the optimization process, leading to $\widehat{\beta}_{\mathcal{D}^*}^M(0) = \widehat{\beta}_{\mathcal{D}}^M(0) = \mathbf{1} \in \mathbb{R}^{dd'}$.

Recall that $X \in \mathbb{R}^{n \times dd'}$, $T_{\pi_\beta} \in \mathbb{R}^n$, $\mathbb{E}[T_{\pi_\beta}] \in \mathbb{R}^n$ and $\widehat{\beta}_{\mathcal{D}}^M$ is $[0,1]^{dd'}$ based on the definition in the original $\ell_0$ optimization problem in equation (22), equation (23) and equation (28). The update rules of using either the dataset $\mathcal{D}$ or $\mathcal{D}^*$ can be written as:

$$\begin{aligned} \widehat{\beta}_{\mathcal{D}}^M(i) &= g_\lambda(\widehat{\beta}_{\mathcal{D}}^M(i-1) + \eta X^T[T_{\pi_\beta} - X\widehat{\beta}_{\mathcal{D}}^M(i-1)]) \\ \beta_{\mathcal{D}^*}^M(i) &= g_\lambda(\widehat{\beta}_{\mathcal{D}^*}^M(i-1) + \eta X^T[\mathbb{E}[T_{\pi_\beta}] - X\widehat{\beta}_{\mathcal{D}^*}^M(i-1)]). \end{aligned} \tag{33}$$

Note that the difference between the two parameters $\widehat{\beta}_{\mathcal{D}}^M$ and $\widehat{\beta}_{\mathcal{D}^*}^M$ essentially relies on the difference between two pairs of transition kernel estimation datasets $[X, \mathbb{E}[T_{\pi_\beta}]]$ and $[X, T_{\pi_\beta}]$. It is easily verified that the hard-thresholding operator $[g_\lambda(\cdot)]_i$ defined in equation (32) is $L = 1$-Lipschitz. According to [90], we can set the learning rate $\eta \leq \frac{1}{L} = 1$ here. Using the Lipschitz property, at each iterative update step $i$, we can control the difference between the estimated parameters obtained by using $\mathcal{D}$ or $\mathcal{D}^*$ as

$$\begin{aligned} &\|\widehat{\beta}_{\mathcal{D}}^M(i) - \widehat{\beta}_{\mathcal{D}^*}^M(i)\|_2 \\ &= \|g_\lambda(\widehat{\beta}_{\mathcal{D}}^M(i-1) + \eta X^T[T_{\pi_\beta} - X\widehat{\beta}_{\mathcal{D}}^M(i-1)]) \\ &\qquad - g_\lambda(\widehat{\beta}_{\mathcal{D}^*}^M(i-1) + \eta X^T[\mathbb{E}[T_{\pi_\beta}] - X\widehat{\beta}_{\mathcal{D}^*}^M(i-1)])\|_2 \\ &\leq \|(\widehat{\beta}_{\mathcal{D}}^M(i-1) + \eta X^T[T_{\pi_\beta} - X\widehat{\beta}_{\mathcal{D}}^M(i-1)]) - (\widehat{\beta}_{\mathcal{D}^*}^M(i-1) + \eta X^T[\mathbb{E}[T_{\pi_\beta}] - X\widehat{\beta}_{\mathcal{D}^*}^M(i-1)])\|_2 \\ &\leq \|(I_{dd'} - \eta X^T X)[\widehat{\beta}_{\mathcal{D}}(i-1) - \widehat{\beta}_{\mathcal{D}^*}(i-1)] + \eta X^T[T_{\pi_\beta} - \mathbb{E}[T_{\pi_\beta}]]\|_2 \\ &\leq \|I_{dd'} - \eta X^T X\|_2 \|\widehat{\beta}_{\mathcal{D}}(i-1) - \widehat{\beta}_{\mathcal{D}^*}(i-1)\|_2 + \eta\|X\|_2\|T_{\pi_\beta} - \mathbb{E}[T_{\pi_\beta}]\|_2 \\ &\leq \|\widehat{\beta}_{\mathcal{D}}(i-1) - \widehat{\beta}_{\mathcal{D}^*}(i-1)\|_2 + \eta\|T_{\pi_\beta} - \mathbb{E}[T_{\pi_\beta}]\|_2, \end{aligned} \tag{34}$$

Here, $I_{dd'}$ represent the identity matrix of size $dd' \times dd'$, the last inequality holds based on the fact that $\|X\|_2 \leq 1$ defined in equation (24). Since $\eta \leq \frac{1}{L} = 1$, as a result, $\|I_{dd'} - \eta X^T X\|_2 = \max(1 - \eta\lambda(X^T X)) \leq 1$. We perform IHT for sufficient $K > 0$ iterations and output the last step estimation as our solutions for either the offline dataset $\mathcal{D}$ (we use for practical optimization) or the perfect dataset $\mathcal{D}^*$. In practice, for any dataset such as $\mathcal{D}$ and any accuracy level $0 \leq \xi \leq 1$, we have the output of IHT gradually converges to the optimal solution $\widehat{\beta}_{\mathcal{D}}^M$ of the problem in equation (26). Namely, after at most $K \simeq \log\left(\frac{\|M\|_0}{\xi}\right)$ steps [89, Corollary 1], the output satisfies $\left\|\widehat{\beta}_{\mathcal{D}}^M - \widehat{\beta}_{\mathcal{D}}^M(K)\right\|_2 \leq \frac{\xi}{4}$. Similarly, based on the perfect dataset $\mathcal{D}^*$, the output of IHT gradually converges to $\widehat{\beta}_{\mathcal{D}^*}^M$ at the same rate. Consequently, the term of interest (a) can be bounded recursively as:

$$\|\widehat{\beta}_{\mathcal{D}}^M - \widehat{\beta}_{\mathcal{D}^*}^M\|_\infty = \left\|\widehat{\beta}_{\mathcal{D}}^M(K) - \widehat{\beta}_{\mathcal{D}^*}^M(K) + \left(\widehat{\beta}_{\mathcal{D}}^M - \widehat{\beta}_{\mathcal{D}}^M(K)\right) + \left(\widehat{\beta}_{\mathcal{D}^*}^M - \widehat{\beta}_{\mathcal{D}^*}^M(K)\right)\right\|_\infty$$

$$\leq \left\|\widehat{\beta}_{\mathcal{D}}^M(K) - \widehat{\beta}_{\mathcal{D}^*}^M(K)\right\|_\infty + \frac{\xi}{4} + \frac{\xi}{4}$$

$$\leq \left\|\widehat{\beta}_{\mathcal{D}}^M(K) - \widehat{\beta}_{\mathcal{D}^*}^M(K)\right\|_2 + \frac{\xi}{4} + \frac{\xi}{4}$$

$$\leq \|\widehat{\beta}_{\mathcal{D}}^M(0) - \widehat{\beta}_{\mathcal{D}^*}^M(0)\|_2 + K\eta\|T_{\pi_\beta} - \mathbb{E}[T_{\pi_\beta}]\|_2 + \frac{\xi}{2}$$

$$= K\eta\|T_{\pi_\beta} - \mathbb{E}[T_{\pi_\beta}]\|_2 + \frac{\xi}{2}$$

$$\lesssim \eta \log\left(\frac{\|M\|_0}{\xi}\right)\|T_{\pi_\beta} - \mathbb{E}[T_{\pi_\beta}]\|_2 + \frac{\xi}{2},$$

(35)

where the second inequality holds by recursively applying equation (34) to $K, K-1, \cdots, 0$ iterations, the last equality is due to the fact that $\widehat{\beta}_{\mathcal{D}}^M(0) = \widehat{\beta}_{\mathcal{D}^*}^M(0)$, and the last inequality holds by setting $K = \log(\frac{\|M\|_2}{\xi})$.

Now the remaining of the proof will focus on controlling $\|T_{\pi_\beta} - \mathbb{E}[T_{\pi_\beta}]\|_\infty$. Recall that we have defined the regression target in equation (23) and equation (28):

$$T_{\pi_\beta}(s'|s,a) = \frac{1}{n(s,a)} \sum_{i=1}^{n(s,a)} \mathbf{1}(s_i = s, a_i = a, s_i' = s'),$$

$$\mathbb{E}[T_{\pi_\beta}(s'|s,a)] = \mathbb{E}_{s' \sim T(\cdot|s,a)}[\mathbf{1}(s_i = s, a_i = a, s_i' = s')].$$

**Proposition 1** (Well-explored dataset). *With the offline dataset $\mathcal{D}$ of in total $n$ samples with $n(s,a)$ samples generated independently conditioned on any $(s,a)$. For any $0 < \delta < 1$, with probability at least $1 - \delta$, one has*

$$\forall (s,a) \in \mathcal{S} \times \mathcal{A}: \quad \|T_{\pi_\beta}(\cdot|s,a) - \mathbb{E}[T_{\pi_\beta}(\cdot|s,a)]\|_2 \leq C_\beta \sqrt{\frac{|\mathcal{S}|}{n(s,a)} \log\left(\frac{|\mathcal{S}||\mathcal{A}|}{\delta}\right)},$$

*for some universal constant $C_\beta$.*

The above proposition can be directly proved by applying [91, Lemma 17] over all $(s,a) \in \mathcal{S} \times \mathcal{A}$:

$$\max_{(s,a) \in \mathcal{S} \times \mathcal{A}} \|T_{\pi_\beta}(\cdot|s,a) - \mathbb{E}[T_{\pi_\beta}(\cdot|s,a)]\|_1 \leq \sqrt{\frac{14|\mathcal{S}|}{n(s,a)} \log\left(\frac{2|\mathcal{S}||\mathcal{A}|}{\delta}\right)}.$$

(36)

As a result, we can further extend the results in equation (35) to bound the term (a) as follows:

$$\|\widehat{\beta}_{\mathcal{D}}^M - \widehat{\beta}_{\mathcal{D}^*}^M\|_\infty \leq K\eta\|T_{\pi_\beta} - \mathbb{E}[T_{\pi_\beta}]\|_2 + \frac{\xi}{2}$$

$$\lesssim \eta C_\beta C_\mu \log\left(\frac{\|M\|_0}{\xi}\right)\sqrt{\frac{|\mathcal{S}|}{n(s,a)} \log\left(\frac{|\mathcal{S}||\mathcal{A}|}{\delta}\right)} + \frac{\xi}{2}$$

$$\leq C_\beta C_\mu \log\left(\frac{\|M\|_0}{\xi}\right)\sqrt{\frac{|\mathcal{S}|}{n(s,a)} \log\left(\frac{|\mathcal{S}||\mathcal{A}|}{\delta}\right)} + \frac{\xi}{2}$$

$$\triangleq C_1 \log\left(\frac{\|M\|_0}{\xi}\right)\sqrt{\frac{|\mathcal{S}|}{n(s,a)} \log\left(\frac{|\mathcal{S}||\mathcal{A}|}{\delta}\right)} + \frac{\xi}{2},$$

(37)

where $C_1 = C_\beta C_\mu$ is some constant that is related to the feature regularity, $\xi$ is the level of error tolerance in the cumulative value returns, in our setting, $0 \leq \xi \leq 1$.

**Step 4: Controlling term (b).** For (b) in Equation (31), which is $\|\widehat{\beta}_{\mathcal{D}^*}^M - \beta^M\|_\infty$, we are interested in what is the optimization error given finite well-explored offline dataset $\mathcal{D}^*$. Here the optimization error mainly originates from the Gaussian noise in the SCM formulation in Definition 4. Yet our causal discovery module, i.e. an $\ell_0$ estimator, will always encounter some estimation error induced by the exogenous noise.

Firstly, we have the bounded relationship between $\ell_2$ and $\ell_\infty$ norm:

$$\|\widehat{\beta}_{\mathcal{D}^*}^M - \beta^M\|_\infty \leq \|\widehat{\beta}_{\mathcal{D}^*}^M - \beta^M\|_2,$$

then we can analyze the error bound for $\ell_0$ regression in the sense of $\ell_2$ norm.

The derivation below generally follows the $\ell_0$ regularized linear regression bound in [92]. Based on Assumption 2 and 3, there exists $M$, we denote this optimal solution in the vector form as $\beta^M$.

Besides the aforementioned optimization error in the iterative thresholding update process, according to the SCM in Definition 4 and Assumption 2 and Proposition 1, we denote a finite subset of observed transition probabilities of the well-explored data $\mathbb{E}[T_{\pi_\beta}]$ with size $n$: $T_{obv} \in \mathbb{R}^n$. By definition above, we can then assume the finite-sample regression target $T$ is generated by causal features of transition pairs (denoted by $X = X_{\pi^*} = X_{\pi_\beta}, X \in \mathbb{R}^{n \times dd'}$) and the ground-truth causal mask (represented by $\beta^M \in \mathbb{R}^{dd'}$) with the following equation:

$$T_{obv} \triangleq \mathbb{E}[T_{\pi_\beta}] = T + \epsilon = X\beta^M + \epsilon. \tag{38}$$

Here $\epsilon \sim \mathcal{N}(0, \sigma I_n)$ is the independent exogenous noise defined in Definition 4. We'll then use the above equation to bound term (b) in equation (31).

Specifically, we solved this $\ell_0$ regression problem with its bounded form as follows:

$$\min_{\beta^M} \|T_{obv} - X\beta^M\|_2^2, \quad s.t. \|\beta^M\|_0 \leq s. \tag{39}$$

In BECAUSE, we select the sparsity level $s \approx \|\beta^M\|_0 = \|M\|_0 \leq dd'$.

For simplicity, we denote the approximate solution $\widehat{\beta}_{\mathcal{D}^*}^M$ in Equation (39) as $\widehat{\beta}^M$. By the virtue of optimality of the solution $\beta^M$ in Equation (39), we find that

$$\|T_{obv} - X\beta^M\|_2^2 \leq \|T_{obv} - X\widehat{\beta}^M\|_2^2, \tag{40}$$

then by expanding and shifting the terms, we can derive the basic inequality for the $\ell_0$ estimator above:

$$\|T_{obv} - X\beta^M\|_2^2 \leq \|(T_{obv} - X\beta^M) + (X\beta^M - X\widehat{\beta}^M)\|_2^2$$
$$\overset{(38)}{\Longrightarrow} \|\epsilon\|_2^2 \leq \|\epsilon + (X\beta^M - X\widehat{\beta}^M)\|_2^2$$
$$= \|\epsilon\|_2^2 + \|X\widehat{\beta}^M - X\beta^M\|_2^2 + 2\langle \epsilon, X\beta^{\beta^*} - X\widehat{\beta}^M \rangle$$

Since both $\widehat{\beta}^M$ and $\beta^M$ are $s$-sparse, the vector $\widehat{\beta}^M - \beta^M$ is at most $2s$-sparse. We denote the set of all $2s$-sparse $dd'$-dimensional vector set as $\mathbb{T}^{dd'}(2s)$, we denote $v = \mathbb{I}(\widehat{\beta}^M - \beta^M = 0)$ as an indicator vector, $v_i = 0$ if and only if $\widehat{\beta}_i^M - \beta_i^M, \forall i \in [dd']$.

By shifting the terms, we use the sub-Gaussian assumption and further use Hölder's inequality to get the following results:

$$\begin{aligned}
\frac{1}{n(s,a)}\|X\widehat{\beta}^M - X\beta^M\|_2^2 &\leq \frac{2}{n(s,a)}\langle X^T\epsilon, \widehat{\beta}^M - \beta^M \rangle \\
&\leq \frac{2}{n(s,a)}\|\widehat{\beta}^M - \beta^M\|_2 \sup_{v \in \mathbb{T}^{dd'}(2s)} \langle v, X^T\epsilon \rangle \\
&\leq 2\|\widehat{\beta}^M - \beta^M\|_2 \sup_{|S|=2s} \|\frac{(X^T\epsilon)_S}{n(s,a)}\|_2 \\
&\leq 2\|\widehat{\beta}^M - \beta^M\|_2 \sigma \sqrt{\frac{2s\log(dd'/\delta)}{n(s,a)}},
\end{aligned} \tag{41}$$

where $\epsilon \sim \mathcal{N}(0, \sigma^2 I_n)$ is the exogenous noise variable in the SCM in Definition 4.

Then by simply applying restricted eigenvalue (RE) condition, for some $\kappa(S) > 0$, we have

$$\kappa(S)\|\widehat{\beta}^M - \beta^M\|_2^2 \leq \frac{1}{n(s,a)}\|X(\widehat{\beta}^M - \beta^M)\|_2^2 \leq 2\|\widehat{\beta}^M - \beta^M\|_2\sigma\sqrt{\frac{2s\log(dd'/\delta)}{n(s,a)}}$$

Therefore, we have:

$$\|\widehat{\beta}^M - \beta^M\|_2^2 \leq \frac{2\sigma\sqrt{2s}}{\kappa(S)}\sqrt{\frac{\log(dd'/\delta)}{n}} \triangleq C_s\sigma\sqrt{\frac{\|M\|_0\log(dd'/\delta)}{n(s,a)}} \tag{42}$$

with probability at least $1 - \delta$, which bounds term (b) in Equation (31).

**Step 5: Summing up the results**   Summarizing both bounds for terms (a) and (b) in Equation (31), we will get the following bounds with probability $1 - \delta$:

$$|(\widehat{\mathbb{B}}_h\widehat{V}_{h+1})(s,a) - (\mathbb{B}_h\widehat{V}_{h+1})(s,a)| \leq \|\widehat{\beta}^M_{\mathcal{D}^*} - \beta^M\|_\infty + \|\widehat{\beta}^M_{\mathcal{D}} - \widehat{\beta}^M_{\mathcal{D}^*}\|_\infty$$

$$\leq C_1 \log\left(\frac{\|M\|_0}{\xi}\right)\sqrt{\frac{|\mathcal{S}|}{n(s,a)}\log\left(\frac{|\mathcal{S}||\mathcal{A}|}{\delta}\right)} + C_s\sigma\sqrt{\|M\|_0}\sqrt{\frac{\log(dd'/\delta)}{n(s,a)}} + \frac{\xi}{2}$$

$$\lesssim \min\left\{C_1 \log\left(\frac{\|M\|_0}{\xi}\right)\sqrt{|\mathcal{S}|}, C_s\sigma\sqrt{\|M\|_0}\right\}\sqrt{\frac{\log(1/\delta)}{n(s,a)}} \tag{43}$$

Therefore, we complete the proof of Lemma 3 by showing that for all $(s,a) \in \mathcal{A} \times \mathcal{A}, h \in [H]$,

$$\mathcal{E}_{\text{BECAUSE}}\left\{|(\widehat{\mathbb{B}}_h\widehat{V}_{h+1})(s,a) - (\mathbb{B}_h\widehat{V}_{h+1})(s,a)|\right.$$

$$\left.\lesssim \min\left\{C_1 \log\left(\frac{\|M\|_0}{\xi}\right)\sqrt{|\mathcal{S}|}, C_s\sigma\sqrt{\|M\|_0}\right\}\sqrt{\frac{\log(1/\delta)}{n(s,a)}}\right\} \tag{44}$$

The final bound of the term (b) includes a dependency of $O(\frac{1}{\sqrt{n}})$ and logarithm of dimensionality $dd'$ in the estimated transition matrix. It also incurs a dependency of error tolerance $\xi$ or SCM's noise level $\sigma$ square root of sparsity level $\sqrt{s} = \sqrt{\|M\|_0}$.

So far, we prove the following bound in theorem 1:

$$V_1^*(\widetilde{s}) - V_1^\pi(\widetilde{s}) \leq \min\left\{C_1 \log\left(\frac{\|M\|_0}{\xi}\right)\sqrt{|\mathcal{S}|}, C_s\sigma\sqrt{\|M\|_0}\right\}\sum_{h=1}^H \mathbb{E}_{\pi^*}\left[\sqrt{\frac{\log(1/\delta)}{n(s_h,a_h)}} \mid s_1 = \widetilde{s}\right], \tag{45}$$

In order to achieve $\xi$-optimal policy such that $V_1^*(\widetilde{s}) - V_1^\pi(\widetilde{s}) \lesssim \xi$, the RHS needs to satisfy:

$$\min\left\{C_1 \log\left(\frac{\|M\|_0}{\xi}\right)\sqrt{|\mathcal{S}|}, C_s\sigma\sqrt{\|M\|_0}\right\}\sum_{h=1}^H \mathbb{E}_{\pi^*}\left[\sqrt{\frac{\log(1/\delta)}{n(s_h,a_h)}} \mid s_1 = \widetilde{s}\right] \lesssim \xi \tag{46}$$

We first multiply $\sqrt{\min_{(s,a,h)\in\mathcal{S}\times\mathcal{A}\times[H]} \mathbb{E}_{\pi^*}[n(s_h,a_h) \mid s_1 = \widetilde{s}]}$, and then take the square on both sides. We have:

$$\min\left\{C_1 \log\left(\frac{\|M\|_0}{\xi}\right)\sqrt{|\mathcal{S}|}, C_s\sigma\sqrt{\|M\|_0}\right\}\sum_{h=1}^H \mathbb{E}_{\pi^*}\left[\sqrt{\frac{\log(1/\delta)\min_{(s,a,h)\in\mathcal{S}\times\mathcal{A}\times[H]} n(s_h,a_h)]}{n(s_h,a_h)}} \mid s_1 = \widetilde{s}\right]$$

$$\lesssim \xi\sqrt{\min_{(s,a,h)\in\mathcal{S}\times\mathcal{A}\times[H]} \mathbb{E}_{\pi^*}[n(s_h,a_h) \mid s_1 = \widetilde{s}]} \tag{47}$$

Given the definition, LHS in Equation (47) satisfies:

$$LHS \lesssim \min\left\{C_1 \log\left(\frac{\|M\|_0}{\xi}\right)\sqrt{|\mathcal{S}|}, C_s\sigma\sqrt{\|M\|_0}\right\} \sum_{h=1}^{H} \mathbb{E}_{\pi^*}\left[\sqrt{\log(1/\delta)} \mid s_1 = \widetilde{s}\right] \quad (48)$$

To ensure the satisfaction of $\xi$-optimal policy, we thus would like the RHS of Equation (47) satisfies:

$$
\begin{aligned}
RHS &= \xi\sqrt{\min_{(s,a,h)\in\mathcal{S}\times\mathcal{A}\times[H]} \mathbb{E}_{\pi^\star}\left[n(s_h, a_h) \mid s_1 = \widetilde{s}\right]} \\
&\gtrsim \min\left\{C_1 \log\left(\frac{\|M\|_0}{\xi}\right)\sqrt{|\mathcal{S}|}, C_s\sigma\sqrt{\|M\|_0}\right\} \sum_{h=1}^{H} \mathbb{E}_{\pi^*}\left[\sqrt{\log(1/\delta)} \mid s_1 = \widetilde{s}\right] \quad (49) \\
&= \min\left\{C_1 \log\left(\frac{\|M\|_0}{\xi}\right)\sqrt{|\mathcal{S}|}, C_s\sigma\sqrt{\|M\|_0}\right\} H\sqrt{\log(1/\delta)}
\end{aligned}
$$

Consequently, we can shift the terms and get the sample complexity bound for the $\xi$-optimal policy, $\forall 0 \leq \xi \leq 1$:

$$\min_{(s,a,h)\in\mathcal{S}\times\mathcal{A}\times[H]} \mathbb{E}_{\pi^\star}\left[n(s_h, a_h) \mid s_1 = \widetilde{s}\right] \gtrsim \frac{\min\left\{C_1^2 \log^2\left(\frac{\|M\|_0}{\xi}\right)|\mathcal{S}|, C_s^2\sigma^2\|M\|_0\right\} \cdot H^2 \log(1/\delta)}{\xi^2}, \quad (50)$$

## C   Additional Experiments Details

In this section, we provide additional experiment results and algorithm implementation details.

### C.1   Implementation of Causal Discovery

We implement the causal discovery primarily based on Equation (5). However, in practice, how to control the coefficient before the sparsity regularization terms is crucial to the final performance. In practice, instead of controlling $\lambda$, we use $p$-value as a threshold to determine the following conditional independence:

$$\phi(s,a)^{(i)} \perp\!\!\!\perp \mu(s')^{(j)} \mid \phi(s,a)^{-(i)}. \quad (51)$$

Here $\phi(\cdot,\cdot)^{(i)}$ means that this element is the $i^{th}$ factor in the abstracted state action representation, similar to $\mu(\cdot,\cdot)^{(j)}$, $\phi(s,a)^{-(i)}$ means all the other factors in the representation except for the $i^{th}$ factor.

If the p-value based on the above conditional independence test is less than a threshold, we can remove the edge by setting $M_{ij} = 0$. Please refer to the Appendix Table 13 for the selection of threshold in each environment.

### C.2   Training Details of Energy-based Model

We train the EBM according to the margin loss in Equation (8). In practice, we attach Tanh() to the output layer to clip the unnormalized score between -1 and +1. The energy networks take in both conditions and samples, then concatenate them together and sent it into MLP encoders. The detailed hyperparameters of EBM are listed in Table 12.

In the vanilla EBM, people follow the Langevin dynamics to effectively sample the negative samples. Here, as we discover the causal mask and identify the causal representation in the model learning stage, we find that we can use these representations in both the energy networks and the sampling process to get some effective negative samples, which is similar to the practice of augmentation of causality-guided counterfactual data [77].

The interesting trick we employ here is the way to get our negative samples by mixing the latent factors from offline data. For example, for a positive sample array

$$\boldsymbol{x}^+ = \begin{bmatrix} \mu(s_1')^{(1)} & \mu(s_1')^{(2)} & \cdots & \mu(s_1')^{(d')} \\ \mu(s_2')^{(1)} & \mu(s_2')^{(2)} & \cdots & \mu(s_2')^{(d')} \\ \cdots & \cdots & \cdots & \cdots \\ \mu(s_n')^{(1)} & \mu(s_n')^{(2)} & \cdots & \mu(s_n')^{(d')} \end{bmatrix}, \tag{52}$$

with conditions:

$$y = \begin{bmatrix} \phi(s_1, a_1)^{(1)} & \phi(s_1, a_1)^{(2)} & \cdots & \phi(s_1, a_1)^{(d)} \\ \phi(s_1, a_1)^{(1)} & \phi(s_1, a_1)^{(2)} & \cdots & \phi(s_1, a_1)^{(d)} \\ \cdots & \cdots & \cdots & \cdots \\ \phi(s_1, a_1)^{(1)} & \phi(s_1, a_1)^{(2)} & \cdots & \phi(s_1, a_1)^{(d)} \end{bmatrix}. \tag{53}$$

Here, $s_i, a_i, s_i'$ denotes the timestep of the offline samples. $\phi(\cdot, \cdot)^{(i)}$ means this element is the $i^{th}$ factor in the abstracted state action representation, similar to the $\mu(\cdot, \cdot)^{(i)}$

as we already get the corresponding causal representation $\mu(s')$ that is semantically meaningful,

we can mix the columns to create useful counterfactual negative samples

$$\boldsymbol{x}^-_{\text{countefactual}} = \begin{bmatrix} \mu(s_1')^{(1)} & \mu(s_2')^{(2)} & \cdots & \mu(s_{n-1}')^{(d')} \\ \mu(s_2')^{(1)} & \mu(s_1')^{(2)} & \cdots & \mu(s_n')^{(d')} \\ \cdots & \cdots & \cdots & \cdots \\ \mu(s_n')^{(1)} & \mu(s_2')^{(2)} & \cdots & \mu(s_1')^{(d')} \end{bmatrix}. \tag{54}$$

These counterfactual negative samples seem to effectively use the causal representation and speed up the training, as we show in Figure 7.

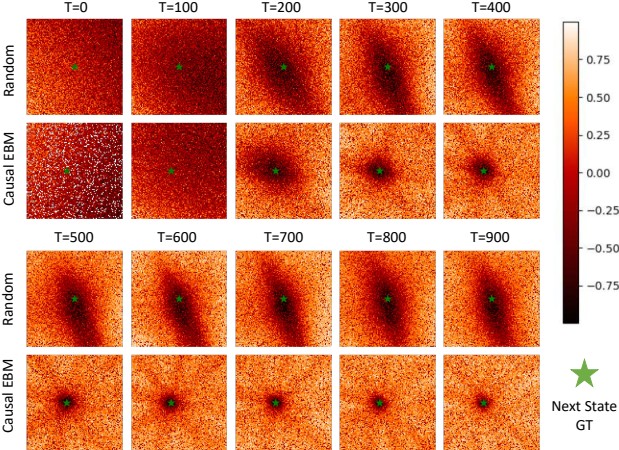

Figure 7: Comparison of the convergence speed in EBM training. Compared to random negative samples, our approach enjoys a higher rate of convergence empirically.

## C.3 Additional Mismatch Analysis

To evaluate the significance of the objective mismatch effect at three different levels of offline datasets in *Unlock* environments, we collect 5,000 episodes in each of the Unlock environments for each method. Then we evaluate the mismatch via the following two metrics.

- We conduct hypothesis testing via Mann-Whitney U Test [93], with Null hypothesis

$$H_0 : \mathcal{L}_{model}(\tau_{pos}) < \mathcal{L}_{model}(\tau_{neg}),$$

- To understand the exact difference in model loss between two groups of samples, we compute their Wasserstein-1 distance in the episodic model loss between the trajectories with positive and negative rewards, i.e. $W_1(\tau_{pos}\|\tau_{neg})$.

We report the results of $p$-value and the $W_1$ distance of two groups of model loss samples in the following table.

Table 5: The comparison results of the $p$-value and the $W_1$ distance ($\times 10^{-4}$) between MOPO and BECAUSE. **Bold** means the better.

| Methods | Unlock-Expert | | Unlock-Medium | | Unlock-Random | |
|---|---|---|---|---|---|---|
| | $p$-value ($\downarrow$) | $W_1$ Dist ($\uparrow$) | $p$-value ($\downarrow$) | $W_1$ Dist ($\uparrow$) | $p$-value ($\downarrow$) | $W_1$ Dist ($\uparrow$) |
| MOPO | $6.5 \times 10^{-5}$ | 0.7 | 1.0 | 1.1 | 1.0 | N.A. |
| Ours | $\approx \mathbf{0}$ | **3.1** | $\approx \mathbf{0}$ | 3.0 | **0.9** | **< 0.1** |

## C.4 Additional Experiment Results

We report the results of the task-wise performance of all baselines in the main experiment and variants in the ablation studies in Table 6, 7, 8, and 9.

Table 6: Success rate (%) for 18 tasks in three different environments. We evaluate the mean and 95% confidence interval given by the $t$-test of the best performance among 10 random seeds, as well as the p-value between the overall performance. **Bold** is the best.

| Env | ICIL | CCIL | TD3+BC | MOPO | GNN | CDL | Denoised | IFactor | MnM | Delphic | Ours |
|---|---|---|---|---|---|---|---|---|---|---|---|
| Lift-I-R | 14.3±9.9 | 10.0±11.4 | 9.7±5.4 | 24.3±2.9 | 22.1±3.6 | **33.8±5.0** | 20.0±4.0 | 24.0±2.8 | 16.3±2.8 | 20.2±3.1 | 19.2±0.6 |
| Lift-O-R | 8.5±4.7 | 0.0±0.0 | 1.3±1.0 | 10.2±1.6 | 13.3±2.3 | 16.0±4.7 | 15.5±3.8 | 21.2±3.0 | 14.2±2.2 | 17.7±2.8 | **21.4±3.9** |
| Unlock-I-R | 3.4±1.1 | 2.2±0.8 | 4.39±0.9 | 21.5±1.9 | 11.7±2.1 | 6.6±0.5 | 6.9±0.7 | 8.1±1.1 | 8.6±1.3 | 14.4±1.1 | **32.7±2.8** |
| Unlock-O-R | 11.6±4.0 | 13.5±3.7 | 13.3±3.0 | 16.6±1.3 | 12.1±1.5 | 7.6±1.0 | 7.0±0.8 | 8.0±1.1 | 9.0±1.0 | 12.2±1.1 | **27.6±2.0** |
| Crash-I-R | 11.4±4.3 | 19.5±10.5 | 9.1±6.6 | 32.7±4.9 | 11.7±0.8 | 39.7±4.6 | 32.0±2.8 | 31.3±4.4 | 16.0±2.2 | 17.4±3.6 | **59.4±6.1** |
| Crash-O-R | 2.8±1.8 | 5.7±3.0 | 0.9±0.6 | 10.0±2.0 | 3.7±0.4 | 10.8±1.9 | 10.8±2.1 | 11.0±3.6 | 4.1±0.6 | 5.4±1.1 | **19.7±1.4** |
| Lift-I-M | 54.0±13.3 | 44.0±20.8 | 26.6±14.8 | 39.8±5.3 | 27.5±5.3 | 32.9±5.7 | 35.3±4.8 | 41.0±5.7 | 28.7±1.8 | 24.0±2.3 | **59.5±4.4** |
| Lift-O-M | **46.8±15.2** | 20.0±21.9 | 13.0±1.1 | 31.9±2.8 | 25.8±1.8 | 26.0±4.2 | 27.0±3.2 | 30.2±2.7 | 24.3±2.3 | 18.3±2.6 | 32.3±4.9 |
| Unlock-I-M | 4.8±0.9 | 4.7±1.3 | 5.4±1.0 | 84.8±5.1 | 17.5±2.6 | 29.7±4.4 | 12.9±1.5 | 37.7±5.5 | 37.6±4.9 | 74.1±1.5 | **98.0±4.9** |
| Unlock-O-M | 12.9±2.8 | 14.1±2.0 | 19.2±2.5 | 39.5±4.7 | 16.2±1.8 | 20.5±3.9 | 10.8±0.8 | 21.5±2.7 | 27.0±3.7 | 51.7±2.3 | **68.8±1.5** |
| Crash-I-M | 35.5±9.9 | 24.5±12.2 | 16.3±9.0 | 47.7±7.3 | 11.5±1.1 | 63.5±4.0 | 63.8±4.0 | 45.5±5.0 | 18.4±2.9 | 58.2±2.2 | **90.4±1.8** |
| Crash-O-M | 11.7±3.5 | 7.9±4.4 | 5.6±3.1 | 17.3±2.3 | 3.8±0.4 | 20.0±2.3 | 20.3±1.6 | 16.0±2.0 | 6.3±1.9 | **22.2±1.7** | 20.3±1.9 |
| Lift-I-E | 73.8±17.0 | 86.7±15.7 | 44.3±12.2 | 82.4±6.7 | 63.3±0.0 | 71.0±5.5 | 74.2±5.5 | **98.0±3.7** | 33.3±3.0 | 53.5±6.3 | 92.8±1.2 |
| Lift-O-E | 54.1±26.1 | 80.0±18.9 | 41.6±16.6 | 72.4±1.9 | 60.8±0.6 | 63.1±5.1 | 64.0±4.4 | 91.7±5.7 | 29.0±4.1 | 49.5±3.1 | **93.7±5.9** |
| Unlock-I-E | 11.6±3.2 | 13.7±3.1 | 15.6±2.2 | 88.8±4.6 | 15.3±1.6 | 73.2±2.8 | 50.3±3.0 | 59.3±2.6 | 60.7±2.2 | 83.2±1.2 | **97.4±1.0** |
| Unlock-O-E | 22.4±5.0 | 35.4±6.0 | 41.6±6.2 | 39.9±4.4 | 13.8±1.4 | 41.3±4.1 | 35.7±2.3 | 29.5±3.5 | 38.7±2.0 | 54.4±1.9 | **82.1±6.5** |
| Crash-I-E | 35.7±9.8 | 26.8±7.2 | 26.0±14.4 | 58.5±4.7 | 11.2±0.9 | 63.7±2.8 | 69.3±3.9 | 52.0±5.3 | 10.2±1.1 | 57.0±1.2 | **95.3±1.3** |
| Crash-O-E | 11.3±3.8 | 12.3±3.5 | 6.2±1.8 | 14.8±2.1 | 3.5±0.9 | 18.8±1.8 | **20.7±2.2** | 15.6±2.3 | 3.9±0.6 | 16.5±1.7 | 20.7±3.2 |
| Overall-I | 27.2 | 25.8 | 17.5 | 53.4 | 21.0 | 44.7 | 40.5 | 44.1 | 25.5 | 44.7 | **73.3** |
| Overall-O | 20.2 | 19.9 | 14.6 | 28.1 | 17.0 | 24.9 | 23.5 | 27.2 | 17.4 | 27.5 | **43.0** |

Table 7: $p$-values of different methods (each has 10 random trials) against BECAUSE in various environments. Under the significance level 0.05, we mark all the baseline results that are significantly lower than BECAUSE as green, and the rest as red. We can see that BECAUSE **significantly** outperforms 10 baselines in 18 tasks in 91.1% of the experiments (164 out of total 180 pairs of experiments).

| Env | ICIL | TD3+BC | MOPO | GNN | CDL | Denoised | IFactor | CCIL | MnM | Delphic |
|---|---|---|---|---|---|---|---|---|---|---|
| Lift-I-random | 0.001 | 0.000 | 0.001 | 0.000 | 0.000 | 0.000 | 0.001 | 0.001 | 0.000 | 0.000 |
| Lift-O-random | 0.000 | 0.000 | 0.000 | 0.001 | 0.033 | 0.013 | 0.464 | 0.000 | 0.001 | 0.052 |
| Unlock-I-random | 0.000 | 0.000 | 0.000 | 0.000 | 0.000 | 0.000 | 0.000 | 0.000 | 0.000 | 0.000 |
| Unlock-O-random | 0.000 | 0.000 | 0.000 | 0.000 | 0.000 | 0.000 | 0.000 | 0.000 | 0.000 | 0.000 |
| Crash-I-random | 0.000 | 0.000 | 0.000 | 0.000 | 0.000 | 0.000 | 0.000 | 0.000 | 0.000 | 0.000 |
| Crash-O-random | 0.000 | 0.000 | 0.000 | 0.000 | 0.000 | 0.000 | 0.000 | 0.000 | 0.000 | 0.000 |
| Lift-I-medium | 0.198 | 0.000 | 0.000 | 0.000 | 0.000 | 0.000 | 0.000 | 0.067 | 0.000 | 0.000 |
| Lift-O-medium | 0.966 | 0.000 | 0.438 | 0.009 | 0.022 | 0.032 | 0.209 | 0.125 | 0.003 | 0.000 |
| Unlock-I-medium | 0.000 | 0.000 | 0.000 | 0.000 | 0.000 | 0.000 | 0.000 | 0.000 | 0.000 | 0.000 |
| Unlock-O-medium | 0.000 | 0.000 | 0.000 | 0.000 | 0.000 | 0.000 | 0.000 | 0.000 | 0.000 | 0.000 |
| Crash-I-medium | 0.000 | 0.000 | 0.000 | 0.000 | 0.000 | 0.000 | 0.000 | 0.000 | 0.000 | 0.000 |
| Crash-O-medium | 0.000 | 0.000 | 0.018 | 0.000 | 0.412 | 0.500 | 0.001 | 0.000 | 0.000 | 0.943 |
| Lift-I-expert | 0.017 | 0.000 | 0.004 | 0.000 | 0.000 | 0.000 | 0.993 | 0.203 | 0.000 | 0.000 |
| Lift-O-expert | 0.004 | 0.000 | 0.000 | 0.000 | 0.000 | 0.000 | 0.297 | 0.027 | 0.000 | 0.000 |
| Unlock-I-expert | 0.000 | 0.000 | 0.001 | 0.000 | 0.000 | 0.000 | 0.000 | 0.000 | 0.000 | 0.000 |
| Unlock-O-expert | 0.000 | 0.000 | 0.000 | 0.000 | 0.000 | 0.000 | 0.000 | 0.000 | 0.000 | 0.000 |
| Crash-I-expert | 0.000 | 0.000 | 0.000 | 0.000 | 0.000 | 0.000 | 0.000 | 0.000 | 0.000 | 0.000 |
| Crash-O-expert | 0.000 | 0.000 | 0.002 | 0.000 | 0.133 | 0.500 | 0.005 | 0.000 | 0.000 | 0.011 |

Table 8: Success rate (%) for 18 tasks in three different environments. We evaluate the mean and 95% confidence interval of the test performance among 10 random seeds. **Bold** means the best.

| Env | **BECAUSE** | BECAUSE-Optimism | BECAUSE-Linear | BECAUSE-Full |
|---|---|---|---|---|
| Lift-I-random | $\mathbf{33.8}_{\pm\mathbf{5.0}}$ | $23.2_{\pm3.1}$ | $16.5_{\pm1.6}$ | $22.2_{\pm6.6}$ |
| Lift-O-random | $\mathbf{21.4}_{\pm\mathbf{3.9}}$ | $15.3_{\pm1.5}$ | $8.9_{\pm4.8}$ | $18.2_{\pm5.3}$ |
| Unlock-I-random | $\mathbf{32.7}_{\pm\mathbf{2.8}}$ | $31.2_{\pm2.4}$ | $20.7_{\pm1.7}$ | $10.7_{\pm0.8}$ |
| Unlock-O-random | $\mathbf{27.6}_{\pm\mathbf{2.1}}$ | $26.3_{\pm1.7}$ | $24.0_{\pm2.7}$ | $9.3_{\pm0.6}$ |
| Crash-I-random | $\mathbf{59.4}_{\pm\mathbf{6.2}}$ | $49.9_{\pm9.2}$ | $54.3_{\pm5.4}$ | $36.1_{\pm7.3}$ |
| Crash-O-random | $\mathbf{19.7}_{\pm\mathbf{1.4}}$ | $14.2_{\pm0.5}$ | $14.8_{\pm1.5}$ | $10.0_{\pm2.2}$ |
| Lift-I-medium | $\mathbf{59.5}_{\pm\mathbf{4.5}}$ | $46.8_{\pm2.1}$ | $24.4_{\pm5.3}$ | $36.4_{\pm6.7}$ |
| Lift-O-medium | $\mathbf{32.3}_{\pm\mathbf{5.0}}$ | $24.5_{\pm2.5}$ | $16.4_{\pm2.3}$ | $28.9_{\pm4.3}$ |
| Unlock-I-medium | $\mathbf{98.0}_{\pm\mathbf{4.9}}$ | $92.7_{\pm5.8}$ | $91.0_{\pm1.8}$ | $29.9_{\pm1.9}$ |
| Unlock-O-medium | $\mathbf{68.8}_{\pm\mathbf{1.5}}$ | $58.7_{\pm2.2}$ | $60.0_{\pm2.0}$ | $18.7_{\pm1.3}$ |
| Crash-I-medium | $\mathbf{90.4}_{\pm\mathbf{1.8}}$ | $82.8_{\pm9.9}$ | $66.7_{\pm7.4}$ | $60.8_{\pm1.8}$ |
| Crash-O-medium | $\mathbf{20.3}_{\pm\mathbf{1.9}}$ | $17.5_{\pm0.0}$ | $15.9_{\pm2.8}$ | $24.6_{\pm0.9}$ |
| Lift-I-expert | $\mathbf{92.8}_{\pm\mathbf{1.2}}$ | $68.6_{\pm4.7}$ | $75.6_{\pm10.7}$ | $78.1_{\pm6.1}$ |
| Lift-O-expert | $\mathbf{93.7}_{\pm\mathbf{6.0}}$ | $58.3_{\pm7.9}$ | $66.1_{\pm8.0}$ | $71.9_{\pm6.9}$ |
| Unlock-I-expert | $\mathbf{97.4}_{\pm\mathbf{1.0}}$ | $93.1_{\pm1.9}$ | $94.0_{\pm2.0}$ | $29.3_{\pm1.3}$ |
| Unlock-O-expert | $\mathbf{82.1}_{\pm\mathbf{6.6}}$ | $64.6_{\pm3.1}$ | $65.8_{\pm3.2}$ | $20.2_{\pm1.7}$ |
| Crash-I-expert | $\mathbf{95.3}_{\pm\mathbf{1.4}}$ | $91.0_{\pm2.2}$ | $77.9_{\pm2.8}$ | $50.3_{\pm7.6}$ |
| Crash-O-expert | $20.7_{\pm3.2}$ | $12.5_{\pm0.0}$ | $\mathbf{26.9}_{\pm\mathbf{6.1}}$ | $25.0_{\pm1.8}$ |
| Overall-I | **73.3** | 64.4 | 57.9 | 39.3 |
| Overall-O | **43.0** | 32.4 | 33.2 | 25.2 |

Table 9: $p$-values of different methods (each has 10 random trials) against BECAUSE in various environments. Under the significance level 0.05, we mark all the baseline results that are significantly lower than BECAUSE as green, and the rest as red. We can see that BECAUSE **significantly** outperforms 3 variants in 18 tasks 83.3% of the experiments (45 out of total 54 pairs of experiments).

| Env | BECAUSE-Optimism | BECAUSE-Linear | BECAUSE-Full |
|---|---|---|---|
| Lift-I-random | 0.001 | 0.000 | 0.003 |
| Lift-O-random | 0.003 | 0.000 | 0.146 |
| Unlock-I-random | 0.189 | 0.000 | 0.000 |
| Unlock-O-random | 0.145 | 0.015 | 0.000 |
| Crash-I-random | 0.036 | 0.090 | 0.000 |
| Crash-O-random | 0.000 | 0.000 | 0.000 |
| Lift-I-medium | 0.000 | 0.000 | 0.000 |
| Lift-O-medium | 0.004 | 0.000 | 0.130 |
| Unlock-I-medium | 0.067 | 0.006 | 0.000 |
| Unlock-O-medium | 0.000 | 0.000 | 0.000 |
| Crash-I-medium | 0.061 | 0.000 | 0.000 |
| Crash-O-medium | 0.005 | 0.005 | 1.000 |
| Lift-I-expert | 0.000 | 0.003 | 0.000 |
| Lift-O-expert | 0.000 | 0.000 | 0.000 |
| Unlock-I-expert | 0.000 | 0.002 | 0.000 |
| Unlock-O-expert | 0.000 | 0.000 | 0.000 |
| Crash-I-expert | 0.001 | 0.000 | 0.000 |
| Crash-O-expert | 0.000 | 0.969 | 0.990 |

## C.5 Additional Environment Description

We provide a more detailed description of the environments we use in the experiment, as shown in Table 10.

**Lift** The Lift environment based on the robosuite [34] contains 33 dimensions of state space, including the end effector pose, joint pose, joint velocity, cube pose as well as its relative position, cube color, and a contact flag. It contains 4 dimensions of hybrid action space that uses Operation Space Control (OSC) to control the 3D position and the 1D gripper movement. The task is counted as a success when the assigned block is lifted from the table over 0.1m. The generalization setting in the Lift environment is to use an unseen combination of position and color during online testing. This environment can be abstracted into 15 dimensions of factorizable state space and 4 dimensions of factorizable action space. The causal graph of this environment is recorded in Figure 8(a).

**Unlock** The Unlock environments based on the MiniGrid world [35] contain 110 dimensions of discrete state space, with 3 of 36-dimensional vector inputs representing the current position of the agent, key, and door in a 6x6 grid world. The rest 2 dimensions in the state space memorize the state of whether the agent has the key in hand. The action space is also discrete (with eight dimensions) to determine the movement (up/down/left/right) and the pick-key, open-door actions. An episode will be counted as a success when the agent holds the key and uses it to open the door in the right position. The generalization setting in the Unlock environment is to change the position of the door and increase the number of total goals in the environment. The agent will only successfully finish one episode by opening all the doors. The causal graph of this environment is recorded in Figure 8(b).

**Crash** The Crash environments are based on the Highway environment [37] which contains 22 dimensions of continuous state space, with four vector inputs representing the current position, velocity, and orientation of the surrounding vehicles and ego vehicles. There are two additional dimensions of state memorizing the collision type between the ego vehicles and surrounding vehicles or pedestrians. The 8-dimensional action space is continuous to determine the acceleration in the $x - y$ directions of the ego and surrounding agents. The generalization of the Crash environment is to add different numbers of pedestrians that may cause the crash. An episode will only end when the ego vehicles have a near-miss with both of the pedestrians at the scene. We visualize the causal graph of this environment in Figure 8(c).

All three environments are visualized in Figure 4. We list their basic configurations in Table 10.

Table 10: Environment configurations used in experiments

| Parameters | Environment | | |
|---|---|---|---|
| | Lift | Unlock | Crash |
| Max step size | 30 | 15 | 30 |
| State dimension | 33 | 110 | 22 |
| Action dimension | 4 | 8 | 8 |
| Action type | Hybrid | Discrete | Hybrid |
| Intrinsic state rank | 15 | 4 | 6 |
| Intrinsic action rank | 4 | 3 | 4 |

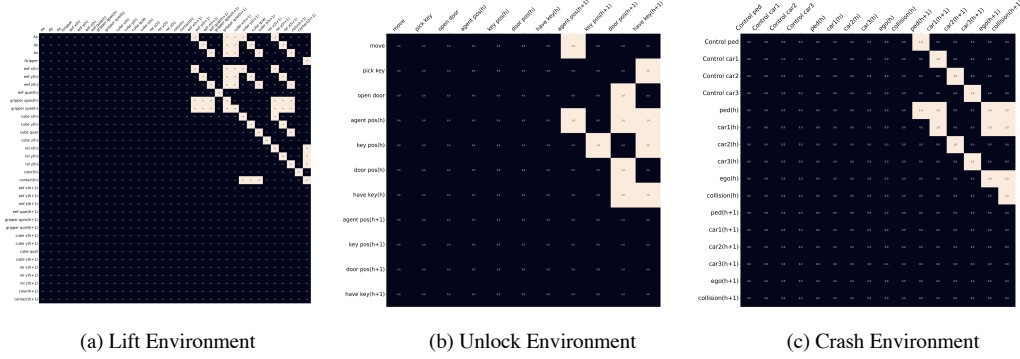

(a) Lift Environment    (b) Unlock Environment    (c) Crash Environment

Figure 8: Underlying causal graph $G$ in all 3 environments with expert demonstration.

## C.6 Additional Baseline Information

We collect data on the above 3 different environments, thus forming 9 groups of offline datasets.

Table 11: Bahavior policies used to collect offline data in different environments.

| Environment | Behavior | #Episodes | Success Rate | Additional Description |
|---|---|---|---|---|
| Lift | Random | 1000 | 0.24 | Random actions *after* a few steps of initialization. |
| | Medium | 1000 | 0.60 | Random actions *before* the goal-reaching expert. |
| | Expert | 1000 | 1.00 | Query expert policy for all time steps. |
| Unlock | Random | 200 | 0.21 | Random navigation with high randomness |
| | Medium | 200 | 0.46 | Targeted searching in goal directions |
| | Expert | 200 | 0.87 | Shortest path planning via $A^*$ |
| Crash | Random | 1000 | 0.14 | Fixed ego, random pedestrians |
| | Medium | 1000 | 0.35 | Planned ego, random pedestrians |
| | Expert | 1000 | 0.66 | Planning in both ego and pedestrians |

After collecting the data using scripted policies in different environments, we train all agents as well as BECAUSE under 10 different random seeds. Then we report the best performance of each trial and compute the mean and standard deviation over 10 seeds for each task in the Appendix 6.

We refer to the following codebase to implement all the baselines we use:

- Invariant Causal Imitation Learning (**ICIL**, [38]): https://github.com/ioanabica/Invariant-Causal-Imitation-Learning, MIT License.
- Causal Confusion Imitation Learning (**CCIL**, [39]): reference link to the paper.
- TD3 with Behavior Cloning (**TD3+BC**, [41]): https://github.com/sfujim/TD3_BC, MIT License.

- Model-based Offline Policy Otimization (**MOPO**, [2]): https://github.com/junming-yang/mopo.git, MIT License.
- Relational Graph Neural Network (**GNN**, [42]): https://github.com/MichSchli/RelationPrediction.git, MIT License.
- Causal Dynamics Learning (**CDL**, [24]): https://github.com/wangzizhao/robosuite/tree/cdl, MIT License.
- Denoised MDP (**Denoised**, [12]): https://github.com/facebookresearch/denoised_mdp.git, CC BY-NC 4.0.
- Mismatch No More (**MnM**, [9]): reference link to the paper.
- World model with identifiable factorization (**IFactor**, [13]), reference link to the paper.
- Delphic Offline RL (**Delphic**, [40]): reference link to the paper.

The detailed hyperparameters we use in BECAUSE and other baselines are listed in Table 12 and Table 13:

## C.7 Experiment Support

Our code is available at the anonymous repo: https://anonymous.4open.science/r/BECAUSE-NeurIPS

**Computing resources** The experiments are run on a server with $2\times$AMD EPYC 7542 32-Core Processor CPU, $2\times$NVIDIA RTX 3090 graphics and $2\times$NVIDIA RTX A6000 graphics, and $252$ GB memory. For one single experiment, it takes BECAUSE and other baselines about $1.5$ hours with $100,000$ iterations to train the world model and $1,000,000$ steps to train the energy-based models.

Table 12: Hyper-parameters of models used in experiments of BECAUSE and baselines (Part I)

| Models | Parameters | Environment | | |
|---|---|---|---|---|
| | | Lift | Unlock | Crash |
| BECAUSE | Learning rate | 0.0001 | 0.001 | 0.0001 |
| | Size of data $\mathcal{D}$ | 15000 | 4000 | 15000 |
| | Epoch per iteration | 20 | 5 | 10 |
| | Batch size | 256 | 256 | 256 |
| | Planning horizon $H$ | 15 | 10 | 20 |
| | Planning population | 1500 | 100 | 1000 |
| | Reward discount $\gamma$ | 0.99 | 0.99 | 0.99 |
| | Spectral norm regularizer $\lambda_\phi$ | $10^{-4}$ | $10^{-4}$ | $10^{-4}$ |
| | Spectral norm regularizer $\lambda_\mu$ | $10^{-4}$ | $10^{-4}$ | $10^{-4}$ |
| | Causal discovery $p_{thres}$ | $10^{-8}$ | $10^{-4}$ | $10^{-6}$ |
| | Encoder hiddens | 256 | 64 | 128 |
| | EBM hidden | 256 | 64 | 128 |
| | EBM negative buffer | 5000 | 1000 | 5000 |
| | EBM training steps | 1000 | 1000 | 1000 |
| | EBM regularizer $\lambda_{\text{EBM}}$ | $10^{-4}$ | $10^{-4}$ | $10^{-4}$ |
| MOPO* | MLP hiddens | 256 | 64 | 128 |
| | MLP layers | 2 | 2 | 2 |
| | Ensemble number | 5 | 5 | 5 |
| CDL* | Initialized mask coef. | 1.0 | 1.0 | 1.0 |
| | MLP hiddens | 256 | 64 | 128 |
| | Sparsity regularizer | 0.001 | 0.001 | 0.001 |
| GNN* | GNN hiddens | 256 | 64 | 128 |
| | GNN layers | 3 | 1 | 3 |

\* Use the same planning parameters as BECAUSE.

Table 13: Hyper-parameters of models used in experiments of baselines (Continued)

| Models | Parameters | Environment | | |
|--------|------------|------|--------|-------|
| | | Lift | Unlock | Crash |
| ICIL | Learning rate of MINE | 0.0001 | 0.0001 | 0.0001 |
| | MINE hiddens | 256 | 64 | 128 |
| | MLP hiddens | 256 | 64 | 128 |
| | Learning rate of EBM | 0.01 | 0.01 | 0.01 |
| | Size of buffer of EBM | 1000 | 1000 | 1000 |
| | EBM training steps | 1000 | 1000 | 1000 |
| | EBM hiddens | 256 | 64 | 128 |
| | K of langevin rollout | 60 | 60 | 60 |
| | $\lambda_{Var}$ of langevin rollout | 0.01 | 0.01 | 0.01 |
| TD3+BC | Learning rate of Critic | 0.0003 | 0.003 | 0.0003 |
| | Critic hiddens | 256 | 64 | 128 |
| | Learning rate of Actor | 0.0003 | 0.001 | 0.0001 |
| | Actor hiddens | 256 | 64 | 128 |
| | Target update rate | 0.005 | 0.001 | 0.0001 |
| | Policy noise | 0.2 | 0.2 | 0.2 |
| | Balance coefficient $\alpha$ | 1.0 | 2.5 | 2.5 |
| Denoised MDP | x belief size | 256 | 64 | 128 |
| | y belief size | 256 | 64 | 128 |
| | z belief size | 0 | 0 | 0 |
| | x state size | 33 | 110 | 22 |
| | y state size | 33 | 110 | 22 |
| | z state size | 0 | 0 | 0 |
| | embedding size | 256 | 64 | 128 |
| | Learning rate | 0.0001 | 0.001 | 0.0001 |
| IFactor | All hidden dim | 256 | 64 | 128 |
| | Disentangled prior output size | 19 | 7 | 10 |
| | Learning rate | 0.0001 | 0.001 | 0.0001 |
| MnM | All hidden dim | 256 | 64 | 128 |
| | Discriminator learning rate | 0.0001 | 0.001 | 0.0001 |
| | Discriminator clip norm | 0.25 | 0.25 | 0.25 |
| CCIL | Reg weight | 0.0001 | 0.001 | 0.0001 |
| | Initial mask probability | 0.95 | 0.95 | 0.95 |
| | Learning rate | 0.0001 | 0.001 | 0.0001 |
| Delphic | Ensemble model size | 5 | 5 | 5 |
| | Uncertainty penalty weight | 0.0001 | 0.0001 | 0.00005 |
| | KL weight | 0.0001 | 0.0001 | 0.00005 |

## C.8 Broader Impact

This work incorporates causality into reinforcement learning methods, which helps humans understand the underlying mechanism of algorithms and check the source of failures. However, the learned causal world model may contain human-readable private information about the environment and the dataset. To mitigate this potential negative societal impact, the causal world model should only be accessible to trustworthy users.

