# OpenReview forum: "BECAUSE: Bilinear Causal Representation for Generalizable Offline Model-based Reinforcement Learning"
_NeurIPS.cc/2024/Conference — NeurIPS 2024 poster_

### Official Review · Reviewer_8enR · 2024-06-22

**Soundness:** 2
**Presentation:** 2
**Contribution:** 2
**Rating:** 2
**Confidence:** 3

**Summary:**

Distribution shifts in the sampling of MBRL will lead to the objective mismatch between model and policy learning.
Therefore, this paper aims to reduce the influence of the distribution shift for MBRL.
It divides the confounders into  $\mu_{\pi}$ and $\mu_{c}$ and then tries to capture causal representations to reduce the influence of the distribution shift, thus mitigating the objective mismatch problem.

**Strengths:**

1. This paper is well-written and easy to follow.
2. In RL, this paper tries to address confounders, which are key problem in causal inference.
3. There are some new theoretical results in this paper.
4. Experiment results illustrate the effectiveness of the proposed method.

**Weaknesses:**

1. The conditions in Bilinear MDPs are difficult to achieve in real-world environments.
2. Some real-world examples can be given for confounders $\mu_{\pi}$ and $\mu_{c}$.

**Questions:**

1. What is the world model in this paper? Does this paper use a world model to train the policy?

---

> ### Author Rebuttal · Authors · 2024-08-07
>
> We would like to express our gratitude to reviewer 8enR for their valuable feedback. The reviewer acknowledges the clarity of our problem formulation and presentation, as well as the theoretical and empirical contribution of this work. We will answer the main questions below:
>
> **Q1. (Generality of Bilinear MDP Formulation): The conditions in Bilinear MDPs are difficult to achieve in real-world environments. Some real-world examples can be given for confounders $u_\pi$ and $u_c$.**
>
> We thank the reviewer for raising these important questions about the generality of bilinear MDP formulation. We discuss two categories of confounders in our experiment, which simulate some real-world problems like indoor navigation, robotic manipulation, and self-driving cars. One real-world example about the two categories of confounders $\mu_\pi$ and $\mu_c$ is in the autonomous driving environments. We have the states ['time-to-collision', 'daytime'] and actions ['throttle', 'steering']. Time-to-collision is the minimum ratio of relative distance to the closest vehicle divided by the velocity. The offline dataset may have two sources of confounders:
> - **Confounders from suboptimal behavior policy:** $u_\pi$ is the confounder introduced by suboptimal behavior policies. For example, one aggressive driver always overtakes the front vehicles. This sub-optimal behavior policy $\pi_\beta(a | s)$ may have a large throttle even when the 'time-to-collision' is extremely small. Such behavior policies introduce confounders that cause spurious correlations between states ('time-to-collision') and actions (throttle). This spurious correlation may cause collision and have low reward if we directly learn from this dataset and deploy it online. In our experiment, we denote different behavior policies as 'random', 'medium', and 'expert' policies, leading to different confounded levels of $u_\pi$.
>
> - **Confounders from environment dynamics**: $u_c$ is the confounder that exists in the transition dynamics $T(s'|s, a)$, and it causes spurious correlation between different state and action components in transition dynamics. In the real-world example, the traffic will be more crowded during the peak hours in a day and less crowded in the rest. Due to the confounder for 'crowdedness' and 'daytime', the model may fail under crowded scenarios out of peak hours, because it will falsely predict a lower 'time-to-collision' outside peak hours even if the traffic is dense. This is because the model is misled by the spurious correlation between 'time-to-collision' in $s'$ and daytime in $s$. In our experiment, we also have different confounders in different tasks in all 3 environments. We have discussed the environments in Appendix C.5.
>
> We visualize the above confounded case in the MDP as **Figure (a) and (b) in the rebuttal supplementary**.
>
> **Q2. (Clarification of World Model): What is the world model in this paper? Does this paper use a world model to train the policy?**
>
> We thank the authors for raising the clarification about the world model. In our setting, our world model is composed of a dynamics model $T(s' | s, a)$ and a reward model $r(s, a)$. More concretely, during the planning phase, we have the following pessimistic value iteration:
> $$
> \overline{Q}(s, a) = r(s, a) - E_\theta(s, a) + \sum_{s'\in \mathcal{S}} \widehat{T}(s'|s, a) \widehat{V}(s').
> $$
>
> Therefore, optimization of our pessimistic policy will be dependent on the learned dynamics model $\widehat{T}(s' | s, a)$ and reward function $r(s, a)$.

---

> > ### Comment · Reviewer_8enR · 2024-08-09
> >
> > World models were first defined in reference [1].
> > Many well-known reinforcement learning algorithms, such as Dreamer [2], are based on the world model.
> > Redefining world models are not appropriate, so I changed my score to 3.
> >
> >
> > [1] David Ha and Jürgen Schmidhuber. World models. arXiv preprint arXiv:1803.10122, 2018.
> > [2] Danijar Hafner, Timothy Lillicrap, Jimmy Ba, and Mohammad Norouzi. Dream to control: Learning behaviors by latent imagination. In International Conference on Learning Representations, 2019.

---

> ### Comment · Reviewer_XKke · 2024-08-09
>
> Dear Reviewer 8enR,
>
> While the reviewer discussions have not officially started, I would like to discuss why the definition of 'world model' presented here is inappropriate. In my opinion, the world model could be defined as the dynamics model and the reward model in the MDP, similar to what the authors have mentioned.
>
> I’m not saying that every aspect of this paper is flawless, but I believe rejecting it solely based on this definition might be a bit unfair. I would appreciate your thoughts on this and welcome any corrections if am wrong.
>
> Reviewer XKke

---

> > ### Comment · Reviewer_8enR · 2024-08-09
> > **Dear Reviewer XKke**
> >
> > Dear Reviewer XKke
> >
> > I think publishing such a definition would mislead many readers, so this is a serious mistake.
> >
> > Reviewer 8enR

---

> > > ### Comment · Reviewer_XKke · 2024-08-09
> > >
> > > Could you please provide further elaboration on which parts are considered 'misleading'? Thank you in advance.

---

> > > > ### Comment · Reviewer_8enR · 2024-08-09
> > > >
> > > > World model (1120 citations) was proposed in 2018 for image-based POMDPs.
> > > > It does not aim to handle MDP problems.
> > > > Many RL methods, such as Dreamer (1228 citations), are based on world models.
> > > > As you can see, the world model in this paper is not for POMDPs.
> > > > Defining such a model with the same name as an existing well-known model will mislead many readers.
> > > > Furthermore, the authors did not cite literature [1] in the paper.
> > > >
> > > >
> > > >
> > > >
> > > >
> > > > [1] David Ha and Jürgen Schmidhuber. World models. arXiv preprint arXiv:1803.10122, 2018.

---

> > > > > ### Author Response · Authors · 2024-08-09
> > > > >
> > > > > We thank reviewer 8enR for engaging in the rebuttal discussion. We also thank reviewer XKke for acknowledging our main contribution. First, we want to clarify that our core contribution is **NOT** proposing a new definition of the world model, but the **bilinear causal representation framework** in model-based RL and causal RL. Then, we address the concern from reviewer 8enR about the "world model" as follows.
> > > > >
> > > > > **'World model' in our paper is not newly defined by us.** Our paper uses a notation of transition model $T(s' | s, a)$ and a reward model $r(s, a)$ to describe the conditional distribution $p(s', r | s, a)$. This "world model" description, **which is not defined by us**, directly follows previous works of model-based RL [3-5], especially two of our baselines Denoised MDP [4] and IFactor [5].
> > > > >
> > > > > **Definition of 'world model' that reviewer 8enR refers to.** To better address the "redefine" concern of the reviewer, we would like to know what mathematical form of "world model" the reviewer 8enR refers to. We summarized the related work [1, 2] mentioned by the reviewer in the following table. We are glad to discuss this question further.
> > > > >
> > > > > **Related works of 'world model'.** Existing papers [1-9] use "world model" with different descriptions due to the broad and controversial meaning of this term. Even [1, 2] mentioned by the reviewer have somewhat different descriptions from each other. And there were works [6-9] proposing this definition that are much earlier than [1, 2]. To have a clear understanding, we summarize the following table of "world models" related to causality or RL topics:
> > > > >
> > > > > |Paper|Description of World Model|
> > > > > |-|-|
> > > > > |[1] Ha, David, and Jürgen Schmidhuber. **"World models."** arXiv preprint arXiv:1803.10122 (2018). | Vision model (V): $p(z_t \| o_t)$ , RNN Memory model (M): $p(z_{t+1} \| a_t, z_t, h_t)$|
> > > > > |[2] Hafner, Danijar, et al.**"Dream to control: Learning behaviors by latent imagination."** arXiv preprint arXiv:1912.01603 (2019). | Representation: $p_\theta(s_{t+1}\|s_t, a_t, o_t)$, transition $q_\theta(s_{t+1}\|s_t, a_t)$, reward: $q_\theta(r\|s)$, action $q_\phi(a_t \| s_t)$, and value $v_\psi(s_t)$|
> > > > > |[3] Li, Minne, et al. **"Causal world models by unsupervised deconfounding of physical dynamics."** arXiv preprint arXiv:2012.14228 (2020).| Conventional **world models**: observational conditional $p(s_{t+1}\|s_t, a_t)$.|
> > > > > |[4] Wang, Tongzhou, et al. **"Denoised MDPs: Learning World Models Better Than the World Itself."** International Conference on Machine Learning. PMLR, 2022.|Latent posterior : $q_\psi(x, y, z\|s, a)$, prediction decoder: $p_\theta(s, r\|x, y, z, a)$.|
> > > > > |[5] Liu, Yuren, et al. **"Learning world models with identifiable factorization."** Advances in Neural Information Processing Systems 36 (2023): 31831-31864. | World model with **disentangled latent dynamics**: Transition model $p(s_{t+1}\|s_t, a_t)$, reward model $p(r_t \| s_t)$, observation model $p(o_t\|s_t)$, representation model $p(s_{t+1}\|o_t, s_t, a_t)$.|
> > > > > |[6] Allen, James F., and Johannes A. Koomen. **"Planning using a temporal world model."** Proceedings of the Eighth international joint conference on Artificial intelligence-Volume 2. 1983.| Temporal logic |
> > > > > |[7] Kaelbling, Leslie Pack, Michael L. Littman, and Andrew W. Moore. **"Reinforcement learning: A survey."** Journal of artificial intelligence research 4 (1996): 237-285.| Locally weighted regression, state estimator |
> > > > > |[8] Wiering, Marco A., and Martijn Van Otterlo. **"Reinforcement learning."** Adaptation, learning, and optimization 12.3 (2012): 729.|Transition models embody knowledge about the environment that can be exploited in various ways.|
> > > > > |[9] Doll, Bradley B., Dylan A. Simon, and Nathaniel D. Daw. **"The ubiquity of model-based reinforcement learning."** Current opinion in neurobiology 22.6 (2012): 1075-1081.| The sequential transition structure of the task (a ‘world model’) |
> > > > >
> > > > > This table shows that prior works have different descriptions of 'world model' in different contexts of problems. In our MDP context, we follow the prior works [3-5] and define our world model with the state transition and reward model. We don't think using our world model under this context misleads readers since a lot of existing work also use the same description.
> > > > >
> > > > > As we mentioned, **our key contribution is the bilinear causal representation learning**. If reviewer 8enR still feels using 'world model' may affect their justification on our contribution, we are happy to add more remarks in our appendix or change 'world model' to other terms that reviewer 8enR feels more appropriate. We believe this change won't have any influence to the core contribution of this paper. We hope the reviewer’s concern will be resolved.

---

> > > > > ### Comment · Reviewer_XKke · 2024-08-10
> > > > > **[Reply to Reviewer 8enR] Regarding the "misleading" part you mentioned**
> > > > >
> > > > > FIrst of all, thank you for your clarification!
> > > > >
> > > > > However, I don't believe the vision component is the key aspect of the "world model". If we assume access to at least some of the grounded symbolic or physical states, then the world model could be primarily concerned with the state dynamics and reward model. These elements essentially characterize the underlying evolution and reward mechanisms in the observations/ measurements within RL. The vision component becomes essential only when the grounded states are not directly observable.

---

> ### Comment · Reviewer_8enR · 2024-08-10
>
> The world model consists of all the  planner's knowledge of the past, present, and future, expressed in a temporal logic [6].
> To optimize POMDP problems,  we should consider not only present state but also the historical information to predict the future states or rewards, therefore world model can be used to optimize POMDP tasks.
>
> However, because of the Markov property, we just need to consider present state in MDPs, which are the key problem in this paper. Can you find any literature, in which world model is for MDPs and does not consider historical information?
>
>
>
>
>
> [6] Allen, James F., and Johannes A. Koomen. "Planning using a temporal world model." Proceedings of the Eighth international joint conference on Artificial intelligence-Volume 2. 1983.

---

> > ### Author Response · Authors · 2024-08-11
> >
> > We thank reviewer 8enR for engaging in the followup discussion. There are a few recent works that use 'world model' in the MDP formulation, including but not limited to [1, 2, 3].
> >
> > Denoised MDPs [1] formulate their problem in MDPs instead of POMDPs in their problem formulation and discuss how the MDP formulation is generic enough for their world model learning problem:
> > > For generality, we consider tasks in the form of Markov Decision Processes (MDPs), described in the usual manner: $\mathcal{M}\triangleq (\mathcal{S}, \mathcal{A}, R, P, p_{s_0})$ (Puterman, 1994), where $\mathcal{S}$ is the state space, $\mathcal{A}$ is the action space, $R$: $\mathcal{S}\to \Delta([0, r_{max}])$ defines the reward random variable $R(s')$ received for arriving at state $s'\in \mathcal{S}$, $P: \mathcal{S} \times \mathcal{A} \to \Delta(\mathcal{S})$ is the transition dynamics, and $p_{s_0} \in \Delta(\mathcal{S})$ defines the distribution of initial state. We use $\Delta(A)$ to denote the set of all distributions over $A$. $P$ and $R$ define the most important components of a MDP: the transition dynamics $P[s'| s, a]$ and the reward function $P[r | s']$. Usually, the objective is to find a policy $\pi: S\to \Delta(A)$ acting based on current state, that maximizes the expected cumulative (discounted) reward.
> > >
> > > Indeed, MDPs provide a general formulation that encompasses many tasks. In fact, the entire real world may be viewed as an MDP with a rich state/observation space S that contains all possible information/signal. For an artificial agent to successfully perform real-world tasks, it must be able to process observations that are incredibly rich and high-dimensional, such as visual or audio signals.
> >
> > As we mentioned in our response, our description of world model follows [1], which is formulated in the MDP setting. We also want to emphasize that **our key contribution is the bilinear causal representation learning**. If reviewer 8enR still feels using 'world model' may affect their justification on our contribution, we are happy to add more remarks in our appendix or **change 'world model' to other terms that reviewer 8enR feels more appropriate**. We believe this change won't have any influence to the core contribution of this paper. Once again, we hope the reviewer’s concern will be resolved, if they can clarify their demands during the discussion phase.
> >
> >
> > > [1] Wang, Tongzhou, et al. "Denoised mdps: Learning world models better than the world itself." arXiv preprint arXiv:2206.15477 (2022).
> > >
> > > [2] Zhu, Zheng-Mao, et al. "Offline reinforcement learning with causal structured world models." arXiv preprint arXiv:2206.01474 (2022).
> > >
> > > [3] Liu, Jingbin, Xinyang Gu, and Shuai Liu. "Reinforcement learning with world model." arXiv preprint arXiv:1908.11494 (2019).

---

> > > ### Comment · Reviewer_8enR · 2024-08-12
> > >
> > > All of [1, 2. 3] take action by considering encoded historcial information.
> > > You may change 'world model' to other terms.

---

> > > > ### Author Response · Authors · 2024-08-12
> > > >
> > > > We think Reviewer 8enR mentioned an important point: **the MDP setting assumes that the current state $s_t$ encoded all history information.** Therefore, $p(s_{t+1}|s_t, a_t)=p(s_{t+1}|s_t, a_t, s_{t-1}, s_{t-2},\cdots)$ already considers all history information prior to $t$ under the Markov assumption. In this context, the 'world model' is applicable in the MDP settings, which is the case in our paper.
> > > >
> > > > However, to avoid further confusion and address the concern of reviewer 8enR, we will change the term 'world model' in our paper to 'dynamics model' (including state transition and reward model). We believe this will address the concern of Reviewer 8enR. Please kindly let us know if there are additional questions or concerns before the discussion period ends. We would appreciate it if Reviewer 8enR would reconsider the evaluation.

---

> ### Comment · Reviewer_8enR · 2024-08-12
>
> **I did not mean that the current state encoded all historical information.**
> In MDPs, the historical information cannot change the probability distributions of $s_{t+1}$, however, with the help of the context in historical information, we may construct a new causal function for MDPs.
> I would like to keep my score.

---

> > ### Comment · Area_Chair_xdUw · 2024-08-13
> >
> > Thanks for the extensive discussion on the term of "World Model". I think it's useful to clarify the definition of the models learned in this paper to avoid potential ambiguity with the term in the literature.
> >
> > Reviewer 8enR, do you have additional major concerns that led to your current rating apart from the use of "world model"?

---

> > > ### Comment · Reviewer_8enR · 2024-08-13
> > >
> > > Yes, I think how to define the model in this paper is an important problem.

---

> > > > ### Author Response · Authors · 2024-08-14
> > > >
> > > > Dear Reviewer 8enR,
> > > >
> > > > Thank you for your response. To help us better address your concern, could you please kindly elaborate more on this sentence "I think how to define the model in this paper is an important problem":
> > > > 1. Which 'model' do you refer to? What is the key issue in our 'definition'?
> > > > 2. How does this issue relate to our core contribution of bilinear causal representation?
> > > >
> > > > If you still feel inappropriate in that we 'redefine' the 'world model' term, we have **changed the term 'world model' to 'dynamics model'**, including the state transition and reward model (see our last response), to avoid further confusion. The term 'dynamics model' is also commonly used by some previous MBRL literature [1, 2].
> > > >
> > > >
> > > > >[1] Janner, Michael, et al. "When to trust your model: Model-based policy optimization." Advances in neural information processing systems 32 (2019).
> > > > >
> > > > >[2] Yu, Tianhe, et al. "Mopo: Model-based offline policy optimization." Advances in Neural Information Processing Systems 33 (2020): 14129-14142.

---

### Official Review · Reviewer_ne9G · 2024-07-04

**Soundness:** 4
**Presentation:** 4
**Contribution:** 4
**Rating:** 7
**Confidence:** 4

**Summary:**

The paper introduces a method to improve model-based offline reinforcement learning by addressing the objective mismatch problem using causal discovery methods. This problem stems from the fact that in this setup the learning algorithm aims to solve the following two problems at once: i) accurate learning of the transition dynamics, ii) identifying the optimal policy on the MDP characterized by the learned transition dynamics. However in practice, an improved solution to one of these two problems does not translate to a better solution to the other one, while in theory it should. The paper's central hypothesis is that this disconnect is caused by the spurious correlations learned by the model which can be blocked by a causality based formulation.

**Strengths:**

* The proposed methodology is very sensible and interesting. Formulating a latent bilinear MDP to enable differentiable causal discovery approaches is a brilliant idea.

* The paper studies the theoretical properties of the presented approach. Although the used theoretical tools are not novel per se, they serve for their purpose excellently. The conceptual connection between causal representation learning and bilinear MDPs is indeed novel and worthwhile acknowledging.

* Despite the denseness of the technical material, the paper is very easy to read and understand.

* The presented empirical results are satisfactory.

**Weaknesses:**

* The compared baselines do make sense. Some of them such as ICIL and CCIL are causal RL approaches. However, the experiments part could be even stronger if the chosen model-based ORL approaches represented the state of the art a little better. TD3+BC and MOPO are outdated. One could for instance consider to compare against a stronger baseline within the same family, such as:

Sun et al., Model-Bellman Inconsistency for Model-based Offline Reinforcement Learning, ICML, 2023.

* It would be interesting to see a demonstration of how well BECAUSE can recover the underlying confounders of a simple MDP. This could be an interesting research question to be investigated in the experiments section. This would enhance the value of the proposed approach from an instrumental to an explanatory one.

**Questions:**

Why was it not possible and/or sensible to study the problem in the most well-known benchmark of offline RL research, the MuJoco environments of D4RL?

**Limitations:**

The paper discusses the limitations of the proposed approach in the final few sentences of the conclusion section. However, it does not discuss the potential negative societal impact of the presented work.

---

> ### Author Rebuttal · Authors · 2024-08-07
>
> We sincerely thank the reviewer for their insightful and inspiring feedback. We are glad to know that the reviewer recognizes the novelty of our contributions, the clarity of our problem formulation, and the empirical results. We answer reviewers' questions about the causal learning process through the bilinear MDP optimization below.
>
> **Q1. (Advanced Model-based Offline RL baselines): The compared baselines do make sense. Some of them such as ICIL and CCIL are causal RL approaches.... consider to compare against a stronger baseline within the same family, such as: (Sun et al., ICML, 2023.).**
>
> We thank the reviewer for proposing this important baseline. We add additional experiments on the *Unlock* environment and illustrate the results in the first table in the general response. While MOBILE indeed demonstrates promising performance in the OOD generalization setting compared to the MOPO baselines, we still observe significant advantage on our methods that learns bilinear causal representation in the world model and use them for planning.
>
> **Q2. (Discussing recovery and explainability of confounders): It would be interesting to see a demonstration of how well BECAUSE can recover the underlying confounders... enhance the value of the proposed approach from an instrumental to an explanatory one.**
>
> As you can see in **Figure (a) and (b) in the rebuttal supplementary**, we give a simple MDP example in the autonomous driving tasks. We have the states ['time-to-collision', 'daytime'] and actions ['throttle', 'steering']. (Time-to-collision (TTC) is the ratio of relative distance to the closest vehicle divided by the velocity, and the TTC gets closer to 0 if a collision happens).
>
> In general, the offline dataset encounters two sources of confounders:
>
> - **Confounders from suboptimal behavior policy:** As is shown in **Figure (a)**, $u_\pi$ is the confounder introduced by suboptimal behavior policies. For example, one aggressive driver always overtakes the front vehicles. This sub-optimal behavior policies $\pi_\beta(a | s)$ may have large throttle even when the 'time-to-collision' is extremely small. Such behavior policies introduce confounders that cause spurious correlations between states ('ttc') and actions ('throttle'). This spurious correlation may cause collision if we directly learn from this dataset and deploy online. In our experiment, we denote different behavior policies as 'random', 'medium', 'expert' policies, leading to different confounded levels of $u_\pi$.
>
> - **Confounders from environment dynamics**: $u_c$ is the confounder that exists in the transition dynamics $T(s'|s, a)$, and it causes spurious correlation between different state and action components in transition dynamics. In the driving example, as shown in **Figure (b)**, the traffic is more crowded (lower ttc) during the peak hours in a day and less crowded (higher ttc) in the rest. With confounder between next 'ttc' and current 'daytime', raw model fails to predict well under crowded scenarios out of peak hours, because it will falsely predict a lower 'ttc' outside peak hours even if the actual traffic is dense. This is because the model is misled by the spurious correlation between 'time-to-collision' in $s'$ and daytime in $s$. In our experiment, we also have different confounders in different tasks in all 3 environments (see Appendix C.5.)
>
> Additionally, we are running some numerical simulations in this simple example, BECAUSE can indeed help identify the transition dynamics. We plan to add this to our final experiment results.
>
> **Q3. (Discussing selected benchmark): Why was it not possible and/or sensible to study the problem in the most well-known benchmark of offline RL research, the MuJoco environments of D4RL?**
>
> Our experiment simulator is adopted by some published works [1, 2, 3] in the causal RL domain. The selected environments have strong complexity in causal structure that leads to distribution mismatch between different training environments. Moreover, they are related to some real-world problems, such as indoor navigation, manipulation, and autonomous vehicles. These tasks all require strong reasoning capability to understand the interaction between the decision-making agents and other static or dynamic environment entities.
> On the other hand, in the mujoco-based D4RL environments, such cause-and-effect structures mainly focus on some physical transformation of the locomotion tasks, which have lower complexity as they do not model the interaction between different entities using the causal structure.
>
> >[1] Wang, Zizhao, et al. "Causal Dynamics Learning for Task-Independent State Abstraction." International Conference on Machine Learning. PMLR, 2022.
> >
> >[2] Ding, Wenhao, et al. "Generalizing goal-conditioned reinforcement learning with variational causal reasoning." Advances in Neural Information Processing Systems 35 (2022): 26532-26548.
> >
> >[3] Ding, Wenhao, et al. "Seeing is not believing: Robust reinforcement learning against spurious correlation." Advances in Neural Information Processing Systems 36 (2023).
>
> **Q4. (Adding societal impact): The paper discusses the limitations of the proposed approach in the final few sentences of the conclusion section. However, it does not discuss the potential negative societal impact of the presented work.**
>
> We thank the authors for raising the concern about potential negative societal impact. We added the following dicussion of societal impacts in our appendix:
>
> *Incorporating causality into model-based reinforcement learning framework provides the explanability of decision making, which helps humans understand the underlying mechanism of algorithms and check the source of failures. However, the learned causal graph may contain human-readable private information about the offline dataset, which could raise privacy issues. To mitigate this potential negative societal impact, the discovered causal graphs should be only accessible to trustworthy users.*

---

> > ### Comment · Reviewer_ne9G · 2024-08-08
> > **Keep score**
> >
> > Thanks for your satisfactory answer. I keep my score.

---

> ### Author Response · Authors · 2024-08-08
>
> We appreciate the dedicated efforts of all the insightful reviews and acknowledgment from reviewer ne9G. The contributive review helps improve our paper's quality during the revision and rebuttal phase.

---

### Official Review · Reviewer_XKke · 2024-07-12

**Soundness:** 2
**Presentation:** 3
**Contribution:** 2
**Rating:** 6
**Confidence:** 3

**Summary:**

This paper aims to learn causal representations in offline RL. The authors propose a novel framework that incorporates causal graph learning through bilinear MDPs to manage mismatches in model learning and policy optimization stages. The framework models two types of confounders: ones affecting state dynamics and those influencing behavior policies. To enhance learning, the framework includes re-weighting and uncertainty quantification as well. Empirical results demonstrate the framework’s effectiveness in both causal discovery for model learning and generalization in offline RL policy learning.

In general, I hold a borderline but positive rating in this initial review. I will consider increasing my rating if the authors could give clarifications on my questions in the following sections (especially questions 1, 2, 3).

**Strengths:**

- [*Motivation and general framework*]: The idea of using causality to model the data generation process in RL (or in general, dynamic systems) gives a principle way to model RL environments. By considering and modeling potential confounders, which often lead to distribution shifts or mismatches, the paper addresses a critical challenge in RL. The decomposition of confounders into dynamics and policy-related aspects is reasonable and well-motivated in this work.

- [*Technical soundness*]: The model and optimization processes are technically robust. The proofs and analyses provided in the appendix have been checked and are generally correct, offering complementary insights to the main part of the paper.

- The writing is clear and easy to follow,

**Weaknesses:**

Since here most of the weaknesses and questions are mixed. So I put both here. One of the major weaknesses, which is also a primary question, is the inherent difficulty in fully learning the causal process through the bilinear MDP optimization (details in question 1-3).

**Question 1**:  [Learning the causal graph by learning $M$] While learning M can partially capture the G graph, two main concerns: (1). Feature space entanglement: In the feature space $\phi$ and $\mu$, the causal variables might be entangled, potentially making the $G$ graph inaccurate (hard to identify from mixed sources). Please clarify if I have any misunderstanding on this; (2) optimization on the search space: Does the optimization search space of $M$ empirically impact the ability to learn the true $G$ graph?

**Question 2**: [About assumption 1] The invariant causal graph assumption simplifies the problem but might be problematic. Specifically: (1). Mapping invariance: $M(u)$ may not be invariant since the scale of confounder effects matters, especially when some latent confounder effects are too small to model or identify in some domains (while in other domains are easy to identify); (2). Deatils on equivalence with assumptions in previous works: Further explanation, preferably with formulas, is needed to clarify the equivalence of this assumption with those in other works (e.g., invariant state representation, action effect) as mentioned in Remark 1.

**Question 3**: [Multiple sources of confounders and identifiability of the causal graph]: How well can M capture different sources of confounders, especially state dynamics confounders $u_c$? How can we ensure the method separates these confounder sources, which is crucial in many RL environments with multiple unobservable latents?

**Question 4**: [Further structure decomposition in world model]: Can the method incorporate algorithms like denoised MDPs to handle cases where state dynamics confounders affect only a subset of states? This could potentially make M more compact.

**Question 5**: [Experimentation with raw pixel inputs]: Can the method work with raw pixel inputs, similar to experiments in works like IFactor cited in the paper? Will it still be effective when learning a visual encoder to extract states?

**Questions:**

I listed all the questions (mixed with weaknesses) in the above section.

**Limitations:**

Limitations have been discussed in the conclusion section. Though there is no in-depth discussion on societal impacts, I don't find any particular societal impacts in this paper compared with other RL works.

---

> ### Author Rebuttal · Authors · 2024-08-07
>
> We would like to express our gratitude to reviewer XKke for their comprehensive and insightful feedback. We are glad to know that the reviewer recognizes the clarity of our problem formulation, the novelty and technical soundness of the proposed method, as well as the theoretical and empirical results. We answer key clarification questions about the causal learning process through the bilinear MDP optimization below.
>
> **Q1. (Learning the causal graph by learning $M$)**:
>
> For the first concern on the feature space entanglement, our learning process alternates between feature learning and causal mask learning, therefore, the causal variables $\phi, \mu$ will best align with the causal relationship. In some harder cases with poor offline data quality (like the *random* behavior policy), we do observe a performance drop and significant objective mismatch issues as is shown in Figure 6. However, our methods still outperform baselines in these cases with the help of bilinear causal representation. Besides, our method is also robust under limited sample size or various spurious level of confounders, as is demonstrated in Figure 4b and 4c.
>
> For the second concern on the optimization on the search space, we'd like to first clarify the relationship between M and G under the formulation of bilinear MDP:
> $$
> G=\\left[\\begin{matrix} 0^{d\\times d} & M \\\\ 0^{d'\\times d} & 0^{d'\\times d'} \\end{matrix}\\right]\\in \\mathbb{R}^{(d+d')\times (d+d')}.
> $$
>
> The optimization of $M$ is a conditional independence testing (CIT) on $d\times d'$ pairs of features. Empirically, our causal discovery process is doing CIT on all the elements in the $d'\times d$ matrix. The optimization search space of $M$ is constrained by the size of $M\in \mathbb{R}^{d'\times d}$. Since the causal discovery is based on hypothesis testing statistics like $\chi^2$ testing, the ability to identify true causal graph $G$ will not be affected by the size of $M$.
>
> **Q2. (About assumption 1)**:
>
> We thank the reviewer for asking insightful questions on our core assumption.
>
> Regarding the first concern on the mapping invariance, if the confounder effects are small as the reviewer mentioned, it means either (a) the confounder is not important to the success of our task in either domain or (b) the dataset we have at hand is too biased to fully learn about true causality.
> - For case (a), we empirically verify the invariance of the discovered causal graph with an experiment on the robustness to the spurious level of confounder, as is mentioned in RQ2 of Section 4.2. In this experiment, we vary the spurious levels (i.e. the scale of confounding effects) by changing the number of confounders in the environments. We show a consistent performance with fewer performance degradation compared to baselines without this bilinear causal representation, as is shown in Figure 4.c. As long as these latent confounders have enough statistical evidence to be identified with causal discovery, we can learn good causal representation.
> - For case (b), which mostly corresponds to the 'random' policy setting where the behavior policy is heavily confounded by the policy confounder $u_\pi$, BECAUSE empirically has better performance compared to other causal RL or MBRL methods.
>
> In conclusion, in either case (a) or (b) raised by the reviewer, BECAUSE can empirically outperform other MBRL baselines, which justifies our assumption of invariant causal mask.
>
> Regarding the second concern on the equivalence with assumptions in previous works, we add some additional discussion in **Table I of our 1-page rebuttal supplementary** with a mathematical description of the specific meaning of the invariant causal graph and how it relates to the assumptions cited in our manuscripts.
>
> **Q3: (Multiple sources of confounders and identifiability of the causal graph)**:
>
> The learning process of $M$ is conducted with two stages: regularized world model learning to deconfound $u_c$ in Equation (5) and (6), and policy-conditioned reweighting formulas to deconfound $u_\pi$ in (7).
> - The first stage is to use the $\ell_0$ objective to regularize the world model loss while maintaining a sparse $M$.
> - In the second stage, we average $M$ by reweighting its expectation across different trajectory batches.
>
> We demonstrate the effect of the above deconfounding process both empirically and theoretically:
> - **Experiment** results show that our method outperforms the existing causal RL and model-based RL baselines in both in-distribution and OOD settings in three environments.
> - **Theoretically**, we provide additional analysis on how well this de-confounding process can identify the causal matrix $M$. Under reasonable assumptions about the offline dataset and underlying dynamics model, we can bound the estimation error of $M$ and have performance guarantees for learned policy in Theorem 1.
>
> **Q4: (Further structure decomposition in world model)**:
>
> We thank the reviewer for providing this inspiring extension of our work. As we discuss in the related works, Denoised MDPs propose a structure decomposition according to the controllability and reward relevance of the state. To ground our bilinear representation in this context, one interesting way is to decompose $\phi, \mu$ into the controllable/non-controllable and reward-relevant/irrelevant parts. This will be an interesting future direction to make $M$ more compact.
>
> **Q5: (Experimentation with raw pixel inputs)**:
>
> To further demonstrate the scalability of our method, we conduct additional experiments with raw-pixel inputs in the *Unlock* environments. The RGB input has a shape of (192, 192, 3), the detailed results can be found in the general response.

---

> > ### Comment · Reviewer_XKke · 2024-08-08
> >
> > Thank you for your detailed response and clarification. Most of my concerns have been well addressed, especially with the newly added Table 1 and the additional results on pixel experiments. I just have one minor follow-up question: for Q3, my initial question was actually about whether maintaining a sparse $M$ here could be the necessary and sufficient condition for estimating the multiple confounders (e.g., different confounders affect different components in the state dynamics). Could the authors provide more insights on this point?
> >
> > For now, given the current quality and the rebuttal, I would increase my rating to 6.

---

> ### Author Response · Authors · 2024-08-08
>
> We thank reviewer XKke for acknowledging our response with additional theoretical and empirical results. We will answer your last clarification question below.
>
> **I just have one minor follow-up question: for Q3, my initial question was actually about whether maintaining a sparse $M$ here could be the necessary and sufficient condition for estimating the multiple confounders (e.g., different confounders affect different components in the state dynamics). Could the authors provide more insights on this point?**
>
> Given our current assumption about the Action State Confounded MDP (ASC-MDP) and Structured Causal Model (SCM), the $\ell_0$ sparsity regularization we apply in the optimization of $M$ will be a **necessary yet not sufficient** condition for estimating all the latent confounders behind the causal graph.
> - The **necessity** of sparsity regularization in $M$ is clear that we need to avoid spurious correlation by 'trimming down' as many unnecessary edges in the transition dynamics while maintaining good prediction accuracy.
> - On the **sufficiency** part, although we cannot guarantee a complete recovery of true causal matrix $M$, we can bound estimation error of $M$ by some polynomial of the following terms:
>     - SCM's noise level $\sigma$ (how strong the confounding effect could be),
>     - Sample size $n$,
>     - Dimension of matrix $d\times d'$,
>     - Structural complexity of ground truth mask $\||M\||_0$
>
>     We have $\text{error}\leq poly(dd', \frac{1}{n}, \sigma, \||M\||_0)$. **Our Appendix B gives a more formal proof of it**. As a result, we can have some high-probability guarantees that if we want a near-optimal ($\epsilon$ error) estimation of the causal matrix $M$, i.e. if we want $\||\widehat{M}-M\||\leq \epsilon$ with high probability $1-\delta$, we may need $n(\epsilon, \delta)$ samples.

---

> ### Comment · Reviewer_XKke · 2024-08-09
>
> Thank you for your further response. My concerns have been well addressed and I would like to keep my positive rating.

---

### Official Review · Reviewer_aSxm · 2024-07-12

**Soundness:** 3
**Presentation:** 3
**Contribution:** 3
**Rating:** 6
**Confidence:** 5

**Summary:**

This paper studies challenges in Model Based rl, namely:
1. Objective mismatch between estimating the model and policy learning for value optimization
2. Confoundedness in offline RL causing this mismatch
The paper proposes a theoretically sound algorithm and experiments extensively to support their claims.

**Strengths:**

1. Well written paper - Problem presentation is very clear
2. Experiments cover a many scenarios, and ablation studies have been provided. I'm curious why multi-task RL algorithms weren't used as baselines given the setup of the experiments.

**Weaknesses:**

1. Has missed some relevant related work particularly in theoretical offline RL which use pessimism based on uncertainty. Please cite :  https://arxiv.org/pdf/2402.12570, https://arxiv.org/pdf/2403.11574
2. The result in Theorem 1 is applicable for a Tabular MDP, but this hasn't been discussed in the paper anywhere. The result becomes void in the case when some (s_h, a_h) along the optimal policy has not been seen in the offline dataset.

**Questions:**

1. How does the Theorem 1 fare to existing works? A detailed comparison on the improvement factors in the sample complexity would be useful.

**Limitations:**

There are no significant limitations of the paper.

---

> ### Author Rebuttal · Authors · 2024-08-07
>
> We would like to express our gratitude to reviewer aSxm for their insightful feedback and acknowledging the novelty of our contributions in practice and theory. We provide our response to the questions below.
>
> **Q1. (Additional Related works): Related works about theoretical offline RL using pessimism... multi-task RL algorithms weren't used as baselines given the setup of the experiments.**
>
> As the reviewer suggested, we added related references to offline RL theory in the related work section.
>
> - **Comparisons with the two related works**: The reviewers raised two relevant works about multi-task RL in offline settings with linear function approximation (linear MDPs). They consider the same sampling mechanism -- offline setting, but different model assumption (linear MDPs) and problem setting (multi-tasks)  as our bilinear MDPs and action-state confounded MDPs.
> - **Differences to multi-task offline RL**: Our work does not primarily target multi-task learning, which aims to solve multiple tasks jointly in an efficient way [2] by leveraging their shared information. In our case, we focus on problems with unobserved causal effects -- the training data has a distribution mismatch with testing environments due to the unobserved confounders. We didn’t include multi-task RL as baselines since they do not target our challenges about causal effects, which makes the comparisons unfair to some extent. So we focus on more related baselines -- 10 causal RL and model-based RL baselines in Section 4.1 and Appendix C.6.
>
> >[1] Jin, Ying, Zhuoran Yang, and Zhaoran Wang. "Is pessimism provably efficient for offline rl?." International Conference on Machine Learning. PMLR, 2021.
> >
> >[2] Yu, Tianhe, et al. "Conservative data sharing for multi-task offline reinforcement learning." Advances in Neural Information Processing Systems 34 (2021): 11501-11516.
>
> **Q2. (Applicability of theoretical results): The result in Theorem 1 is applicable for a Tabular MDP, but this hasn't been discussed in the paper anywhere.**
>
> The reviewer is correct that Theorem 1 is primarily applicable for the tabular case when the state and action space are finite. We added a remark under the theorem as the reviewer suggested:
> - **Our Theorem can be extended to continuous state space.** Theorem 1 provides the performance guarantees of our proposed method in tabular case with finite state and action space. It lays a solid foundation to carry out more general cases with continuous state and action space. The current results can be extended to continuous state space by using covering number argument in our key lemmas, while handling action space need more adaptation and we leave it as interesting future works.
> - **Continuous space is potentially reduced to tabular cases in practice.** Our proposed algorithm can be extended to continuous state or action space via some additional derivation on feature extraction. Prior works such as VQ-VAE [1] provide powerful encoders that can tokenize a continuous visual input into some discrete latent space with finite elements. We will add more discussion on the point in the revised manuscripts.
>
> **Q3. (Theoretical results justification): The result becomes void in the case when some (s_h, a_h) along the optimal policy has not been seen in the offline dataset.**
>
> The reviewer is insightful that offline dataset **do** need to satisfy some properties to make offline learning possible and well-posed. To our best knowledge, the current weakest assumption for the offline dataset to ensure provably learning an optimal policy is called (clipped) single-policy concentrability assumption [2, 3] (such as Definition 1 in [3]) -- the offline dataset need to include sufficient data over the region (state-action tuples $(s_h, a_h)$) that the optimal policy can visit. Under such weakest assumption, our Theorem 1 can ensure the performance of the proposed algorithm.
>
> >[1] Van Den Oord, Aaron, and Oriol Vinyals. "Neural discrete representation learning." Advances in neural information processing systems 30 (2017).
> >
> >[2] Li, Gen, et al. "Settling the sample complexity of model-based offline reinforcement learning." The Annals of Statistics 52.1 (2024): 233-260.
> >
> >[3] Rashidinejad, Paria, et al. "Bridging offline reinforcement learning and imitation learning: A tale of pessimism." Advances in Neural Information Processing Systems 34 (2021): 11702-11716.
>
>
> **Q4. (Comparison with existing theoretical works): How does the Theorem 1 fare to existing works? A detailed comparison on the improvement factors in the sample complexity would be useful.**
>
> We thank the reviewer for raising this question -- which is our key theoretical technical contribution (the proof pipeline for Theorem 1). **Please refer to the general response for details due to the limited space.** We provide a shortened version here:
>
> - **Technical contributions: developed new proof pipeline.** We target a new problem --- bilinear MDPs with sparse core matrix $M$ --- brings new challenges since we need to use iterative algorithms with no closed-form solution, while prior works usually heavily depend on having a closed-form solution from the optimization problem. We develop a new proof pipeline that considers the accumulative errors during the process that we believe is of independent interest for future RL works.
> - **Sample complexity comparisons with prior works.** To our knowledge, we are the first to focus on the problem  —- bilinear MDPs with a sparse core matrix $M$ for causal deconfounding and generalization. So our sample complexity results can't be directly compared to existing works. But using prior works as references, it verifies the tightness of our results --- having a comparable dependency on all the salient parameters ($H$, $d$, $\xi$).

---

> > ### Comment · Reviewer_aSxm · 2024-08-07
> >
> > i thank the authors for their detailed responses, as well as adding the relevant papers in the related work. I've increased my score.

---

> > > ### Author Response · Authors · 2024-08-08
> > >
> > > We appreciate the dedicated efforts of all the insightful reviews from reviewer aSxm. The contributive review helps improve our paper's quality during the revision and rebuttal phase.

---

### Official Review · Reviewer_8KHa · 2024-07-20

**Soundness:** 4
**Presentation:** 4
**Contribution:** 4
**Rating:** 7
**Confidence:** 4

**Summary:**

The paper proposes BECAUSE, an algorithm designed to address the objective mismatch between model and policy learning in offline model-based reinforcement learning (MBRL). The algorithm first models the spurious correlations between the current state s and the current action a, as well as between the next state s' and the state-action pair (s, a), through the formulation of action-state confounded Markov Decision Processes (ASC-MDP). Based on this formulation, a compact method for learning causal representations is introduced. Once these causal representations are acquired, they are employed in both world model learning and planning to enhance the algorithm’s robustness and generalizability.

**Strengths:**

The paper is well-written, proposing their intuition backed by three clearly laid out steps to walk the readers through the working of the algorithm. The insight of mitigates the objective mismatch with causal awareness learned from offline data is novel. The paper also provides a thorough theoretical analysis, including proofs of error bounds and sample efficiency.

**Weaknesses:**

The method relies on certain assumptions, such as the invariance of causal graphs, which might not always hold in real-world scenarios. Additionally, modeling the causal relationships with a linear structure may be overly simplistic, potentially limiting the accuracy and effectiveness of the algorithm in capturing more complex dependencies.

**Questions:**

Is this method efficient in handling continuous actions? During the planning phase, computing the argmax Q can be time-consuming.
Limitation: The proposed method introduces additional complexity to the MBRL framework, which may pose challenges for practical implementation and scalability.

---

> ### Author Rebuttal · Authors · 2024-08-07
>
> We sincerely thank reviewer 8KHa for their insightful and inspiring feedback. We are glad to know that the reviewer recognizes the novelty of our contributions, the clarity of our problem formulation, the theoretical contributions, and the empirical evidence that shows the advantages compared to baselines. We provide our response to the questions below.
>
> **Q1. (Assumption on Invariant Causal Graph and Bilinear Structure): The method relies on certain assumptions, such as the invariance of causal graphs, which might not always hold in real-world scenarios. Additionally, modeling the causal relationships with a linear structure may be overly simplistic, potentially limiting the accuracy and effectiveness of the algorithm in capturing more complex dependencies.**
>
> We thank the reviewer for this insightful discussion on our assumption. We elaborate more on the generality of invariant causal graph assumption and the bilinear representation structures.
> 1. Our assumption of invariant causal graph targets on the generalizable RL setting when the training and testing environments have distribution shift but share the underlying **cause-and-effect relationship**. This assumption is commonly made in many prior causal RL works, such as [1, 2, 3]. This causal invariance setting is actually applicable in many real-world decision-makin tasks like indoor navigation, autonomous driving and manipulation, as is shown in our experiment.
> 2. Regarding the over-simplification concern about our bilinear structure in MDP, the nonlinearity of the transition dynamics can be captured by the feature encoder $\phi(s, a), \mu(s')$. In more scalable contexts such as the foundation model, linear structure is also commonly used in CLIP [4], where the inner product is applied in the latent space.
>
>
> >[1] Zhang, Amy, et al. "Invariant causal prediction for block mdps." International Conference on Machine Learning. PMLR, 2020.
> >
> >[2] Wang, Zizhao, et al. "Causal Dynamics Learning for Task-Independent State Abstraction." International Conference on Machine Learning. PMLR, 2022.
> >
> >[3] Zhu, Wenxuan, Chao Yu, and Qiang Zhang. "Causal deep reinforcement learning using observational data." Proceedings of the Thirty-Second International Joint Conference on Artificial Intelligence. 2023.
> >
> >[4] Radford, Alec, et al. "Learning transferable visual models from natural language supervision." International conference on machine learning. PMLR, 2021.
>
> **Q2. (Continuous action): Is this method efficient in handling continuous actions? During the planning phase, computing the argmax Q can be time-consuming.**
>
> We thank the reviewer for asking this important question in implementing the planner. As the reviewer mentioned about the planning phase, our method theoretically uses the following pessimistic value iteration:
> $$
> \overline{Q}(s, a) = r(s, a) - E_\theta(s, a) + \sum_{s'\in \mathcal{S}} \widehat{T}(s'|s, a) \widehat{V}(s').
> $$
> In practice, the above formulation is applicable to continuous actions we use model predictive control (MPC). We first uniformly sample random actions in the action space $a_{sample}\in \mathcal{A}$, then roll out future states by imagination using the learned dynamics model $\hat{T}(s'|s, a_{\text{sample}})$. Finally, we will have a set of Q functions $\overline{Q}(s, a_{\text{sample}})$ and take $a = \arg\max_{a_{\text{sample}}\in \mathcal{A}} \overline{Q}(s, a_{\text{sample}})$ as the selected actions to rollout the next step.
>
> We denote the number of random action samples per step as "*planning population*", which is a fixed hyperparameter for all the methods in experiment environments in Appendix Table 9.
>
> **Q3. (Scalability): The proposed method introduces additional complexity to the MBRL framework, which may pose challenges for practical implementation and scalability.**
>
> We agree that our method adds additional structure on top of traditional MBRL in exchange of better OOD generalizability. To further demonstrate the scalability and computational efficiency of our method, we conduct additional experiments with **raw-pixel inputs in the Unlock environments** (see supplementary Figure (c) for details). The RGB input has a shape of (192, 192, 3) and we encode them with 3-layer CNN. The results in the **second table of the general response/Table III in the supplementary** show that BECAUSE still outperforms ICIL and IFactor baselines with visual inputs by a clear margin.
> We also evaluate the inference speed of our method with both visual inputs and attach details in the one-page PDF. We believe the inference speed (Table IV in supplementary) is still in the acceptable range with a comparable inference speed as IFactor (>20 FPS on image data).
>
> In conclusion, the empirical evidence verifies (a) the scalability of BECAUSE and (b) the computational efficiency with high-dimensional inputs.

---

### Author Rebuttal · Authors · 2024-08-07

## **General Response**

We thank all the reviewers for their dedicated efforts in providing valuable feedback to our work. All of the reviewers acknowledge our clarity of problem formulation and the technical soundness in bridging bilinear MDP with causality.
In this general response, we first clarify some important technical contributions, then we attach two sets of new experiment results: (1) a comparison with an advanced MBRL baseline, and (2) another experiment with raw pixel inputs.

### **Theoretical comparison with existing works**: for both technical contributions and final sample complexity results
Recall that we target bilinear MDP in the offline setting with the transition kernel $T = \phi^T M \mu$, where the core matrix $M$ is assumed to be sparse and unknown (represent the causal graph).

- **New challenges in our problems: no closed-form optimization solutions.** We aim to solve bilinear MDPs with sparse core matrix $M$, which leads to a $\ell_0$ regression problem that is usually solved by iterative algorithms without a closed-form solution. Such a closed-form situation brings daunting challenges for developing provable suboptimality guarantees since we can’t explicitly write the error term to be controlled out anymore. Existing proof pipelines in prior works for (bi)-linear MDPs can’t be applied here since they heavily depend on having a closed-form optimization solution  [1, 2, 3].
- **Our new proof pipeline for non-closed form optimization problems.**  Prior works using ridge regression only need to control the statistical error of the final closed-form output without focusing on the optimization process. While for our $\ell_0$-regularized optimization, as no closed-form solution can be given, we consider the accumulative errors throughout the optimization process, summing errors from each iteration to determine the final error terms. This new proof pipeline potentially generalizes to a wide range of RL problems without closed-form solutions, which we believe is of independent interest to RL theory, especially for RL algorithms involving iterative methods like Lasso.
- **Sample complexity comparisons with prior works.** First, we would like to highlight that we target a different problem  —- bilinear MDPs with a sparse core matrix $M$ for causal deconfounding and generalization. To our knowledge, we are the first to focus on this problem, so our sample complexity results can't be directly compared to existing works. As suggested by the reviewer, we will include the sample complexity results of existing works targeting bilinear MDPs [1-3] as references, verifying the tightness of our results.  Specifically, we denote the size of the latent space as $d$ and the planning horizon as $H$, for any $\xi$-optimal policy, the sample complexity in [2, 3] is bounded with $\mathcal{O}(\frac{H^7 d^2 \log(H^2d)}{\xi^2} \log^2(\frac{Hd}{\xi}))$ and $\mathcal{O}(\frac{H^4d}{\xi^2})$, respectively.  In our work, this sample complexity is bounded with a rate of $\mathcal{O}(\frac{\|M\|_0 H^2 \log (d)}{\xi^2 } \log^2(\frac{\|M\|_0}{\xi}))$, where $\|M\|_0\ll d^2$ is the sparsity level of core matrix $M$. Our work has a comparable dependency on all the salient parameters (H, d,$\xi$) compared with prior works.

>[1] Yang, Lin, and Mengdi Wang. "Reinforcement learning in feature space: Matrix bandit, kernels, and regret bound." International Conference on Machine Learning. PMLR, 2020.
>
>[2] Du, Simon, et al. "Bilinear classes: A structural framework for provable generalization in rl." International Conference on Machine Learning. PMLR, 2021.
>
>[3] Zhang, Weitong, et al. "Provably efficient representation selection in low-rank Markov decision processes: from online to offline RL." Uncertainty in Artificial Intelligence. PMLR, 2023.

### **Additional experiments**


We illustrate the success rate (%) comparison between **MOBILE**, **MOPO** and **Ours** (BECAUSE) in the *Unlock* environments with six evaluation settings. We run experiments with 10 random seeds and report the mean and 95% confidence interval with t-testing.
We can see some improvements of MOBILE over MOPO under some OOD settings, yet the gap between MOBILE and our methods is still significant.

|Env|MOPO|MOBILE|Ours|Env|MOPO|MOBILE|Ours|
|-|-|-|-|-|-|-|-|
|Unlock-I-R|21.5(1.9)| 15.9(1.0) | **32.7(2.8)** | Unlock-O-R | 16.6 (1.3) | 12.8(0.8) | **27.6(2.0)** |
|Unlock-I-M|84.8(5.1)| 72.4(1.7) | **98.0(4.9)** | Unlock-O-M | 39.5(4.7)| 40.7(1.8) | **68.8(1.5)** |
|Unlock-I-E|88.8(4.6)| 78.3(1.2) | **97.4(1.0)** | Unlock-O-E | 39.9(4.4) | 45.6(2.1) | **82.1(6.5)** |

----------

We experiment on **RGB images** instead of vector observation of *Unlock*, and compare our method with two of our baselines, ICIL (model-free) and IFactor (model-based). We run experiments with 10 random seeds and report the mean success rate (%) and its 95% confidence interval with t-testing.  All the methods have performance decay compared to the vector state setting, yet our method can still prevail under visual inputs:
|Env | ICIL | IFactor | Ours | Env | ICIL | IFactor | Ours |
|-|-|-|-|-|-|-|-|
|Unlock-I-R|0.8(0.8) | 4.3(1.1) | **15.7(3.3)** | Unlock-O-R | 1.5(1.8) | 4.7(1.6) | **5.9(0.9)** |
|Unlock-I-M|5.3(2.0) | 30.2(4.1) | **62.0(4.6)** | Unlock-O-M | 8.6(4.2) | 15.4(2.4) | **71.6(9.1)** |
|Unlock-I-E|8.7(3.4) | 34.0(4.8) | **63.7(3.9)** | Unlock-O-E | 17.1(4.2) | 16.7(3.1) | **73.6(19.5)** |

In addition to the above general response, we also illustrate some tables that discuss related work, additional experiment details and new illustration examples in the one-page supplementary material.

---

### Decision · Program_Chairs · 2024-09-25

**Decision:**

Accept (poster)

**Comment:**

This paper studies the objective mismatch in offline model-based RL between learning the transition dynamics and the optimal policy of the MDP. Improving one objective does not necessarily improve the other in practice. It hypothesizes the disconnection is caused by the spurious correlations and proposes to solve the problem with a bilinear causal representation.

**Pros**
- Most reviewers consider the paper is well written and easy to follow
- The method is well motivated and the proposed method is technically sound
- Thorough theoretical analysis
- Experiments and ablation study show the benefit of the proposed method

**Cons**
- Multiple reviewers were concerned on the assumption of invariant caual graph and bilinear causal structure. The authors provided detailed discussion on its applicability in real problems.
- Reviewer aSxm pointed out some missing related work, which authors have provided additional disucssion.
- Reviewer ne9G was concerned of a lack of experiments with stronger baselines. The authors provided more experiments accordingly.
- Reviewer 8enR is concerned of the use of term "world model" that led to a lengthy discussion. While there's still disagreement on whether its appropriate to use "world model", it does not impact the core contribution of the this submission.

I would strongly encourage the authors to incorporate their additional remarks and clarification in the rebuttal in their final response. Please also make a clear definition of the "world model" or dynamics model, if they decide to rename it to, in the revision to avoid ambiguities from the literature.